# BlurDM: A Blur Diffusion Model for Image Deblurring

**Jin-Ting He**[1]     **Fu-Jen Tsai**[2]     **Yan-Tsung Peng**[3]     **Min-Hung Chen**[4]
**Chia-Wen Lin**[2]     **Yen-Yu Lin**[1]
[1]National Yang Ming Chiao Tung University     [2]National Tsing Hua University
[3]National Chengchi University     [4]NVIDIA
jinting.cs12@nycu.edu.tw   fjtsai@gapp.nthu.edu.tw   ytpeng@cs.nccu.edu.tw
minhungc@nvidia.com   cwlin@ee.nthu.edu.tw   lin@cs.nycu.edu.tw

## Abstract

Diffusion models show promise for dynamic scene deblurring; however, existing studies often fail to leverage the intrinsic nature of the blurring process within diffusion models, limiting their full potential. To address it, we present a Blur Diffusion Model (BlurDM), which seamlessly integrates the blur formation process into diffusion for image deblurring. Observing that motion blur stems from continuous exposure, BlurDM implicitly models the blur formation process through a dual-diffusion forward scheme, diffusing both noise and blur onto a sharp image. During the reverse generation process, we derive a dual denoising and deblurring formulation, enabling BlurDM to recover the sharp image by simultaneously denoising and deblurring, given pure Gaussian noise conditioned on the blurred image as input. Additionally, to efficiently integrate BlurDM into deblurring networks, we perform BlurDM in the latent space, forming a flexible prior generation network for deblurring. Extensive experiments demonstrate that BlurDM significantly and consistently enhances existing deblurring methods on four benchmark datasets. The project page is available at https://jin-ting-he.github.io/BlurDM/.

## 1   Introduction

Camera shake or moving objects frequently introduce unwanted blur artifacts in captured images, severely degrading image quality and hindering downstream vision applications, such as object detection [11, 41], semantic segmentation [1, 43], and face recognition [12, 23]. Dynamic scene image deblurring aims to restore sharp details from a single blurred image, a highly ill-posed problem due to the directional and non-uniform nature of blur.

With the advancement of deep learning, CNN-based models [6, 24, 26, 39, 50, 52] have demonstrated remarkable success in data-driven deblurring. Additionally, Transformer-based approaches [2, 13, 21, 40, 42, 49] have been introduced to effectively capture long-range dependencies, further enhancing deblurring performance by leveraging global contextual information. Although previous methods have successfully improved deblurring performance, the inherent constraints of regression loss [4] typically lead to over-smoothed results with limited high-frequency details.

Recent advances in diffusion models [10, 32, 37] have demonstrated remarkable success in image generation, producing high-quality images with rich details and sharp textures through a forward noise diffusion followed by reverse denoising. Building on the success of diffusion models, several studies [4, 17, 28, 29, 47] have incorporated them into deblurring models to produce restored images. However, standard diffusion models are not specifically designed for deblurring. Thus, directly applying them to deblurring networks limits their potential, leading to suboptimal performance.

39th Conference on Neural Information Processing Systems (NeurIPS 2025).

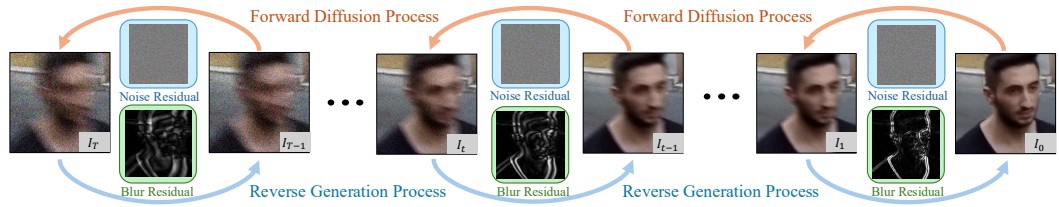

Figure 1: BlurDM is a diffusion-based network that leverages inductive bias of blur formation for dynamic scene deblurring. It progressively adds noise and blur in the forward process and iteratively estimate and removes them in the reverse process to recover sharp images.

The limitation arises from a fundamental discrepancy between the diffusion process and the motion blur formation process. Unlike random noise in the standard diffusion process, motion blur results from a continuous exposure process during image capture, where blur intensity accumulates progressively along a motion trajectory. As a result, motion blur exhibits structured and directed patterns, rather than the random noise perturbations modeled in conventional diffusion processes. To bridge this gap, we propose a diffusion model that mimics the physical formation of motion blur, highlighting the continuous and progressive characteristics of blur formation. Instead of solely relying on standard noise diffusion, our approach incorporates a blur diffusion mechanism, which gradually introduces motion blur to sharp images in a structured manner. By leveraging the iterative nature of diffusion models, the proposed framework integrates the inductive bias of continuous blur formation, enhancing its ability to recover fine details and preserve image structures.

In this paper, we propose the Blur Diffusion Model (BlurDM), a novel approach that aligns the diffusion process with the physical principles of blur formation to enhance deblurring performance. BlurDM adopts a dual diffusion strategy, combining noise and blur diffusion to truly reflect the progressive nature of blur formation, as illustrated in Fig.1. In the forward diffusion process, BlurDM progressively adds both noise and blur to sharp images to generate blurred and noisy images. To achieve a gradual increase in blur, it is crucial to gauge the blur residual, representing the incremental blur added as the exposure time extends. However, existing deblurring datasets primarily consist of blurred-sharp image pairs without ground-truth blur residuals. To address this, BlurDM employs a continuous blur accumulation formulation to implicitly represent the blur residual without relying on ground-truth blur residuals. This enables BlurDM to gradually blur images to align with the principles of the blur formation process.

In the reverse generation process, BlurDM aims to simultaneously remove noise and blur to restore sharp images. To overcome the challenge of unavailable ground-truth blur residuals, we derive a dual denoising and deblurring formulation that follows the principles of the blur formation process to implicitly approximate noise and blur residuals through dedicated noise and blur residual estimators. By effectively reversing the blur formation process, BlurDM can generate high-quality, realistic, sharp images. However, when applied to image deblurring, diffusion models may struggle to accurately reconstruct details with high content fidelity due to their inherent stochastic nature [48]. To address the limitations, inspired by [4, 47], we utilize BlurDM as a prior generation network to flexibly and efficiently enhance existing deblurring models. Guided by the priors learned via BlurDM, deblurred models can achieve more accurate and visually consistent results. Key contributions of this work are summarized as follows:

- We present BlurDM, a novel diffusion-based network that incorporates the inductive bias of blur formation to enhance dynamic scene image deblurring.

- We propose a dual noise and blur diffusion process and derive a dual denoising and deblurring formulation, which allows BlurDM to implicitly estimate blur residuals instead of relying on the ground truth.

- Extensive experiments demonstrate that BlurDM significantly and consistently improves four deblurring models on four benchmark datasets.

## 2 Related Work

### 2.1 Image Deblurring

Image deblurring has made substantial progress with the development of deep learning. Numerous studies have explored CNN-based deblurring using recurrent architectures, such as multi-scale [24,

39], multi-patch [51, 52], and multi-temporal [26] recurrent networks. For example, Tau et al. [39] develop a scale-recurrent network accompanied by a coarse-to-fine strategy for deblurring. Zamir et al. [51] introduce a multi-stage patch-recurrent network that splits an image into non-overlapping patches for hierarchical blurred pattern handling. Park et al. [26] designs a temporal-recurrent network that progressively recovers sharp images through incremental temporal training.

Transformer-based methods [2, 13, 21, 40, 42, 49] have recently garnered considerable attention for deblurring due to their capacities to model long-range dependencies. However, the substantial training data and memory requirements of Transformers motivate the development of efficient variants [13, 21, 40, 42, 49] specifically tailored for deblurring. For instance, Zamir et al. [49] introduced a channel-wise attention mechanism to reduce memory overhead. Tsai et al. [40] proposed a strip-wise attention mechanism to handle blurred patterns with diverse orientations and magnitudes. Kong et al. [13] presented frequency attention to replace dot product operations in the spatial domain with element-wise multiplications in the frequency domain. Mao et al. [21] incorporated local channel-wise attention in the frequency domain to capture cross-covariance in the attention mechanism.

Although the aforementioned advances have improved deblurring performance through various architectural and algorithmic designs, the inherent constraints of using the regression loss [4] frequently lead to over-smoothed results with limited high-frequency details, producing suboptimal deblurred images.

## 2.2 Diffusion Models

Diffusion models [10, 38] have demonstrated remarkable capability in generating high-fidelity images with rich details through forward noise diffusion and reverse denoising. They have been leveraged in numerous studies [20, 22, 25, 27, 33, 34, 45, 46, 53] to synthesize high-quality images under a variety of conditioning schemes.

Diffusion models have been applied to low-level vision tasks [7, 9, 14, 18, 19, 47, 54]. For instance, Xia et al. [47] employed diffusion models to generate prior representations for clean image recovery. Liu et al. [19] utilized text prompts to compile task-specific priors across various image restoration tasks. Zheng et al. [54] proposed a selective hourglass mapping strategy to learn shared information between different tasks for universal image restoration.

Recognizing the advances of diffusion models in low-level vision, researchers have extended their use to image deblurring [3, 4, 15, 16, 17, 28, 30, 44]. Specifically, Whang et al. [44] introduced a stochastic refinement diffusion model for deblurring. Ren et al. [30] incorporated a multi-scale structure guidance network within the diffusion model to recover sharp images. Furthermore, several studies [3, 4, 16, 28] employed diffusion models as prior generation networks and perform diffusion in the latent space to improve deblurring efficiency. For instance, Chen et al. [4] proposed a hierarchical integration module to fuse diffusion priors for deblurring, while Chen et al. [3] incorporated these priors into window-based transformer blocks. While these methods effectively reduce diffusion model latency for deblurring, they overlook the intrinsic characteristics of the blurring process within the diffusion framework, limiting their full potential.

Although Liu et al. [17] proposed residual diffusion by computing the difference between sharp and blurred images using a subtraction operation, the blur formation process is inherently a convolutional process rather than a direct additive difference, making this approach insufficient for accurately capturing blur characteristics. In contrast, we propose a novel framework that incorporates the blur formation process into the diffusion model, leading to significant deblurring performance improvements.

## 3 Proposed Method

We propose Blur Diffusion Model (BlurDM), a novel diffusion framework for image deblurring. Unlike existing methods [4, 28, 29, 47], which rely only on noise diffusion, BlurDM integrates a blur diffusion process, incorporating blur formation into diffusion to improve deblurring performance.

As shown in Fig. 1, we progressively add both noise and blur to a sharp image through a dual noise and blur diffusion process during forward diffusion. In the reverse process, BlurDM jointly denoises

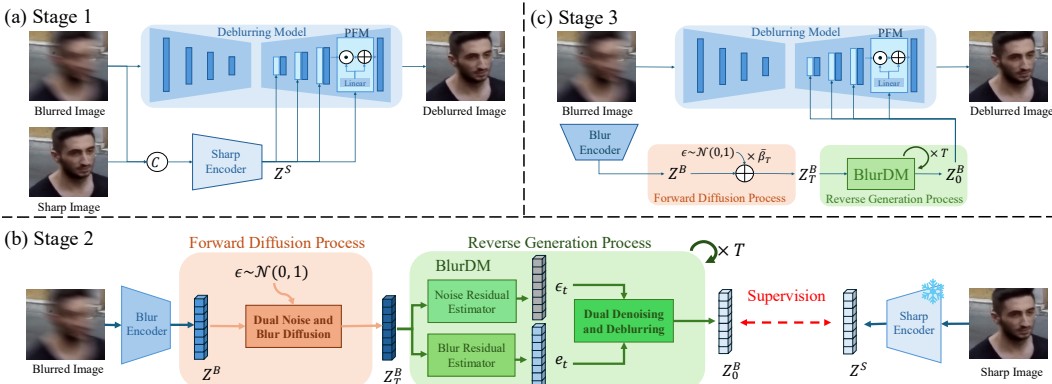

Figure 2: Overall framework of the proposed method. (a) Stage 1: Pre-train the Sharp Encoder (SE), Prior Fusion Module (PFM), and the deblurring network to obtain the sharp prior $\tilde{Z}^S$. (b) Stage 2: Optimize the Blur Encoder (BE) and BlurDM to learn the diffusion prior $Z_0^B$ from a blurred image. (c) Stage 3: Jointly optimize the BE, PFM, BlurDM, and deblurring network to generate the final deblurred image.

and deblurs the image, starting from Gaussian noise conditioned on the blurred input. Ultimately, we use BlurDM as a prior generation network to retain the diffusion model's ability to learn high-quality, realistic image content while embedding the learned prior into the latent space of a deblurring network for effective and high-fidelity restoration, as shown in Fig. 2. Next, we detail the key components of BlurDM, including the dual noise and blur diffusion process, dual denoising and deblurring formulation, and network architecture.

## 3.1 Dual Noise and Blur Diffusion Process

Motion blur in the image capture process is introduced from continuous exposure, where the camera sensors accumulate light over the exposure duration, causing the blending of moving elements along the motion trajectories and leading to a gradual build-up in blur. This process can be mathematically represented as $B = \frac{1}{\alpha_T} \int_{\tau=0}^{\alpha_T} H(\tau) \, d\tau$, where $B \in \mathbb{R}^{H \times W \times 3}$, $H(\tau) \in \mathbb{R}^{H \times W \times 3}$, and $\alpha_T$ denote the blurred image, the instantaneous scene radiance at each moment $\tau$, and the total exposure time, respectively. This models the blur formation process, showing how continuous light integration during exposure results in the accumulation of motion blur. Building on the understanding of the blur formation process, we propose a dual-diffusion framework that incorporates the blur formation framework into noise diffusion. That is, a sharp image is progressively corrupted by both noise and blur in the forward diffusion, capturing the concept of blur degradation introduced during continuous exposure.

To differentiate between images captured at varying exposure periods, we define a sharp and clean image $I_0$, obtained within a short, proper exposure time $\alpha_0$ ($\alpha_0 < \alpha_T$), as $I_0 = \frac{1}{\alpha_0} \int_{\tau=0}^{\alpha_0} H(\tau) \, d\tau$, where $I_0 \in \mathbb{R}^{H \times W \times 3}$ represents the sharp image captured with minimal blur. The contrast between $B$ and $I_0$ indicates the effect of exposure duration on motion blur, meaning longer exposure $\alpha_T$ introduces more blur, whereas a proper exposure $\alpha_0$ yields a sharp image. Our objective is to progressively add noise and blur to $I_0$ based on the blur formation process. The dual noise and blur diffusion process at the next time step can be defined as

$$I_1 = \frac{1}{\alpha_1} \int_{\tau=0}^{\alpha_1} H(\tau) \, d\tau + \beta_1 \epsilon_1 = \frac{1}{\alpha_1} \left( \int_{\tau=0}^{\alpha_0} H(\tau) d\tau + \int_{\tau=\alpha_0}^{\alpha_1} H(\tau) d\tau \right) + \beta_1 \epsilon_1 \qquad (1)$$

$$= \frac{\alpha_0}{\alpha_1} I_0 + \frac{1}{\alpha_1} \int_{\tau=\alpha_0}^{\alpha_1} H(\tau) d\tau + \beta_1 \epsilon_1 = \frac{\alpha_0}{\alpha_1} I_0 + \frac{1}{\alpha_1} e_1 + \beta_1 \epsilon_1, \qquad (2)$$

where $I_1$ represents the intermediate blurred and noisy image, corresponding to the exposure period from 0 to $\alpha_1$ ($\alpha_0 < \alpha_1 < \alpha_T$), $\epsilon_1$ is pure Gaussian noise, $\beta_1$ denotes the noise scaling coefficient, and $e_1 = \int_{\tau=\alpha_0}^{\alpha_1} H(\tau) d\tau$ is the blur residual that accumulates from $\alpha_0$ to $\alpha_1$. Based on (2), the forward transition at time $t$ is defined as

$$I_t = \frac{\alpha_{t-1}}{\alpha_t} I_{t-1} + \frac{1}{\alpha_t} e_t + \beta_t \epsilon_t, \qquad (3)$$

where $\epsilon_t \sim \mathcal{N}(0, \mathbf{I})$ and $e_t = \int_{\tau=\alpha_{t-1}}^{\alpha_t} H(\tau)d\tau$ denotes the blur residual accumulating from $\alpha_{t-1}$ to $\alpha_t$ during the exposure process.

From (3), each forward step from $I_{t-1}$ to $I_t$ is a Gaussian transition, where the blur residual $e_t$ introduces a deterministic mean shift to the distribution. Specifically, the transition distribution is

$$q(I_t \mid I_{t-1}, e_t) = \mathcal{N}\left(I_t; \frac{\alpha_{t-1}}{\alpha_t}I_{t-1} + \frac{1}{\alpha_t}e_t, \beta_t^2\mathbf{I}\right). \tag{4}$$

By iterating (4), we generate a sequence of progressively blurred and noisy images $\{I_1, I_2, \ldots, I_T\}$ through a $T$-step diffusion process. The complete forward sampling probability is therefore given by

$$q(I_{1:T} \mid I_0, e_{1:T}) = \prod_{t=1}^{T} q(I_t \mid I_{t-1}, e_t). \tag{5}$$

However, existing deblur datasets typically consist of blurry-sharp image pairs without providing the corresponding blur residuals. To address this limitation, we reparameterize (5) to obtain the conditional probability distribution $q(I_T|I_0, e_{1:T})$ [10], as shown in (6). The full derivation is provided in Appendix A.1.

$$q(I_T|I_0, e_{1:T}) = \mathcal{N}\left(I_T; \frac{\alpha_0}{\alpha_T}I_0 + \frac{1}{\alpha_T}\sum_{t=1}^{T}e_t, \bar{\beta}_T^2\mathbf{I}\right), \quad \text{where } \bar{\beta}_T = \left(\sqrt{\sum_{t=1}^{T}\left(\frac{\alpha_t}{\alpha_T}\right)^2\beta_t^2}\right). \tag{6}$$

The final blurred and noisy image $I_T$ can be sampled from the distribution $q(I_T|I_0, e_{1:T})$ as

$$I_T = \frac{\alpha_0}{\alpha_T}I_0 + \frac{1}{\alpha_T}\sum_{t=1}^{T}e_t + \bar{\beta}_T\epsilon = \frac{1}{\alpha_T}\int_{\tau=0}^{\alpha_T} H(\tau)\,d\tau + \bar{\beta}_T\epsilon = B + \bar{\beta}_T\epsilon, \tag{7}$$

where the final blurred and noisy image $I_T$ can be generated in a single step by adding noise to the input blurred image $B$. This formulation preserves the Gaussian nature of the diffusion process while embedding the blur information directly into the mean of the distribution through a physically grounded shift. Next, we detail the dual denoising and deblurring formulation for the reverse generation process.

### 3.2 Dual Denoising and Deblurring Process

In the reverse generation process, we aim to progressively remove both noise and blur from the degraded image $I_T$ to recover the sharp image $I_0$, based on our dual denoising and deblurring framework. Unlike standard diffusion models that start from pure Gaussian noise, our method samples the terminal observation $I_T$ from a Gaussian distribution $\mathcal{N}(I_T; B, \bar{\beta}_T^2\mathbf{I})$, where $B$ is a fully blurred input image. To reconstruct $I_0$, we use a blur residual estimator $e^\theta(I_t, t, B)$ and a noise estimator $\epsilon^\theta(I_t, t, B)$ to approximate the respective components, $e_t$ and $\epsilon_t$, at each step.

Inspired by the deterministic sampling formulation in DDIM [38], we define the reverse transition distribution as

$$p_\theta(I_{t-1} \mid I_t) = q_\sigma(I_{t-1} \mid I_t, I_0^\theta, e_{1:t}^\theta), \quad \text{where} \quad I_0^\theta = \frac{\alpha_t}{\alpha_0}I_t - \frac{1}{\alpha_0}\sum_{i=1}^{t}e_i^\theta - \frac{\alpha_t}{\alpha_0}\bar{\beta}_t\epsilon^\theta. \tag{8}$$

The transition probability $q_\sigma$ is defined as

$$q_\sigma(I_{t-1} \mid I_t, I_0^\theta, e_{1:t}^\theta) \tag{9}$$
$$= \mathcal{N}\left(I_{t-1}; \frac{\alpha_0}{\alpha_{t-1}}I_0^\theta + \frac{1}{\alpha_{t-1}}\sum_{i=1}^{t-1}e_i^\theta + \sqrt{\bar{\beta}_{t-1}^2 - \sigma_t^2} \cdot \frac{I_t - \left(\frac{\alpha_0}{\alpha_t}I_0^\theta + \frac{1}{\alpha_t}\sum_{i=1}^{t}e_i^\theta\right)}{\bar{\beta}_t}, \sigma_t^2\mathbf{I}\right),$$

with variance term $\sigma_t^2 = \eta \cdot \frac{\beta_t^2 \bar{\beta}_{t-1}^2}{\bar{\beta}_t^2}$. When $\eta = 0$, this yields a deterministic sampling, which simplifies the reverse step to

$$I_{t-1} = \frac{\alpha_t}{\alpha_{t-1}}I_t - \frac{1}{\alpha_{t-1}}e^\theta(I_t, t, B) - \left(\frac{\alpha_t\bar{\beta}_t}{\alpha_{t-1}} - \bar{\beta}_{t-1}\right)\epsilon^\theta(I_t, t, B). \tag{10}$$

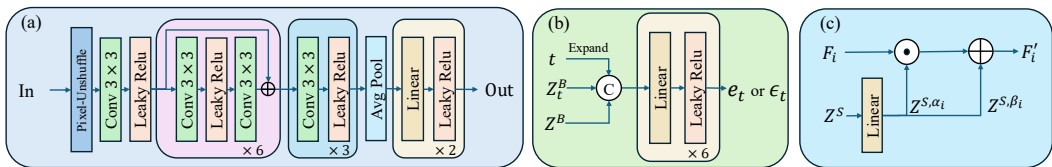

Figure 3: Architecture of the Sharp/Blur Encoders (a), Blur/Noise Estimators (b), and the Prior Fusion Module (c).

Complete derivations of the variational lower bound, optimization objectives, and sampling formulation are provided in Appendix A.2 and Appendix A.3. In the following, we detail the optimization of the blur residual estimator $e^\theta$ and the noise estimator $\epsilon^\theta$ in the latent space for BlurDM.

### 3.3 Latent BlurDM

To efficiently integrate BlurDM into deblurring networks, we develop it in the latent space, where it serves as a prior generator to enhance existing deblurring methods. Inspired by previous work [4, 47], we adopt a three-stage training strategy for effective integration, as illustrated in Fig. 2. This process guides the latent features to capture physically meaningful representations for blur residuals, allowing the model to encode exposure-aware information in the latent space.

**First Stage.** We begin by pre-training the deblurring networks with the Sharp Encoder (SE) and the Prior Fusion Module (PFM). Specifically, given a blurred image $B \in \mathbb{R}^{H \times W \times 3}$ and its sharp counterpart $S \in \mathbb{R}^{H \times W \times 3}$, we concatenate them to feed into the SE to obtain the sharp prior as $Z^S = \mathbf{SE}(\mathbf{Concate}(B, S)) \in \mathbb{R}^{1 \times 1 \times C}$. Subsequently, we fuse $Z^S$ with the decoder features $F_i \in \mathbb{R}^{h_i \times w_i \times c_i}$ at each scale of a deblurring network using PFM, generating the fused features $F'_i$ of the $i$-th scale. Specifically, PFM generates the affine parameters $Z^{S,\alpha_i} \in \mathbb{R}^{1 \times 1 \times c_i}$ and $Z^{S,\beta_i} \in \mathbb{R}^{1 \times 1 \times c_i}$ from $Z^S$ by a linear transformation, and modulate $F$ as

$$(Z^{S,\alpha_i}, Z^{S,\beta_i}) = \mathbf{Linear}(Z^S), \quad F'_i = Z^{S,\alpha_i} \times F_i + Z^{S,\beta_i}, \tag{11}$$

where $\times$ and $+$ denote channel-wise multiplication and addition, respectively. Thus, we can generate a deblurred image $O \in \mathbb{R}^{H \times W \times 3}$, enhanced by the sharp prior $Z^S$, by supervising $O$ with the sharp image $S$.

**Second Stage.** Since sharp images are unavailable during testing, we estimate the sharp prior $Z^S$ from the blurred image $B$ using the proposed BlurDM, treating $Z^S$ as the ground-truth prior at this stage. To achieve this, we employ a Blur Encoder (BE), structurally identical to SE, to generate $Z^B \in \mathbb{R}^{1 \times 1 \times C}$ from $B$. Next, we introduce noise into $Z^B$ following (6) to obtain $Z^B_T$, defined as $Z^B_T = Z^B + \bar{\beta}_T \epsilon$, aligning the diffusion process with blur formation. Finally, we iteratively remove both noise and blur from $Z^B_T$ using (10) to generate the diffusion prior $Z^B_0$ via

$$Z^B_{t-1} = \frac{\alpha_t}{\alpha_{t-1}} Z^B_t - \frac{1}{\alpha_{t-1}} e^\theta(Z^B_t, t, Z^B) - (\frac{\alpha_t \bar{\beta}_t}{\alpha_{t-1}} - \bar{\beta}_{t-1}) \epsilon^\theta(Z^B_t, t, Z^B), \tag{12}$$

which is used to estimate the sharp prior $Z^S$. Recent studies [4, 8, 35, 47] reveal that supervision on the final output can effectively influence the entire diffusion trajectory. We define a latent-prior loss $\mathcal{L}_{\text{prior}} = \left\| Z^B_0 - Z^S \right\|_1$, where $Z^B_0$ is obtained by recursively removing estimated blur and noise residuals from $I_T$ via the shared estimators $e^\theta$ and $\epsilon^\theta$. By back-propagating $\mathcal{L}_{\text{prior}}$, gradients are distributed across all reverse steps, furnishing amortised trajectory-level supervision without step-wise labels. Further details are provided in Appendix A.2.

**Third Stage.** We jointly optimize the pre-trained BE, BlurDM, PFM, and deblurring networks from the first and second stages to generate the deblurred image $O$, ensuring that the learned diffusion prior $Z^B_0$ effectively enhances deblurring performance. We supervise $O$ with $S$ using the loss function originally designed for the given deblurring network. After this stage, the final model we obtain consists of BE, BlurDM, and the deblurring network for inference. We further provide the theoretical justification of latent BlurDM in Appendix A.4.

Table 1: Quantitative results on GoPro, HIDE, RealBlur-J, and RealBlur-R datasets, where "Baseline" and "BlurDM" denote the image deblurring performances without and with BlurDM, respectively. Arrows indicate the direction of improvement (PSNR↑, SSIM↑, LPIPS↓).

| Method | | GoPro PSNR↑ | SSIM↑ | LPIPS↓ | HIDE PSNR↑ | SSIM↑ | LPIPS↓ | RealBlur-J PSNR↑ | SSIM↑ | LPIPS↓ | RealBlur-R PSNR↑ | SSIM↑ | LPIPS↓ |
|---|---|---|---|---|---|---|---|---|---|---|---|---|---|
| MIMO-UNet | Baseline | 32.44 | 0.957 | 0.0115 | 30.00 | 0.930 | 0.0217 | 31.59 | 0.918 | 0.0345 | 39.03 | 0.968 | 0.0215 |
| | BlurDM | **32.93** | **0.961** | **0.0091** | **30.73** | **0.939** | **0.0168** | **32.13** | **0.926** | **0.0264** | **39.63** | **0.972** | **0.0172** |
| Stripformer | Baseline | 33.09 | 0.962 | 0.0085 | 31.03 | 0.940 | 0.0147 | 32.48 | 0.929 | 0.0222 | 39.84 | 0.974 | 0.0138 |
| | BlurDM | **33.53** | **0.966** | **0.0074** | **31.36** | **0.944** | **0.0122** | **33.53** | **0.938** | **0.0175** | **41.00** | **0.977** | **0.0115** |
| FFTformer | Baseline | 34.21 | 0.969 | 0.0067 | 31.62 | 0.946 | 0.0153 | 32.62 | 0.933 | 0.0220 | 40.11 | 0.973 | 0.0149 |
| | BlurDM | **34.34** | **0.970** | **0.0060** | **31.76** | **0.947** | **0.0145** | **32.92** | **0.939** | **0.0195** | **40.55** | **0.975** | **0.0136** |
| LoFormer | Baseline | 33.54 | 0.966 | 0.0084 | 31.18 | 0.943 | 0.0176 | 32.23 | 0.932 | 0.0223 | 40.36 | 0.974 | 0.0148 |
| | BlurDM | **33.70** | **0.967** | **0.0073** | **31.27** | **0.944** | **0.0158** | **33.47** | **0.941** | **0.0189** | **40.92** | **0.976** | **0.0127** |
| **Average Gain** | | **+0.31** | **+0.003** | **-0.0013** | **+0.32** | **+0.004** | **-0.0025** | **+0.78** | **+0.008** | **-0.0047** | **+0.69** | **+0.003** | **-0.0025** |

# 4 Experiments

## 4.1 Experimental Setup

**Implementation Details.** Fig. 3 illustrates the architectural design of the four components in BlurDM: the Sharp Encoder (SE), Blur Encoder (BE), BlurDM, and Prior Fusion Module (PFM). Specifically, SE and BE have the same network architecture, each with six residual blocks, four CNN layers, and two MLP layers. BlurDM contains noise and blur residual estimators, each comprising six MLP layers. PFM consists of one MLP layer. We empirically set $T = 5$ in BlurDM, with $\beta_{1:T}$ increasing uniformly from 0 to 0.02 and $\alpha_{0:T}$ increasing uniformly from 0 to 1. The overall framework (Third Stage) is optimized using the default training settings of each deblurring model, including learning rate, number of epochs, batch size, optimizer, etc., to ensure fair comparisons.

**Deblurring Models and Datasets.** We adopt four prominent deblurring models, including MIMO-UNet [5], Stripformer [40], FFTformer [13], and LoFormer [21], to validate the effectiveness of BlurDM. Following previous work [5, 13, 21, 40], we adopt the widely used GoPro [24] and HIDE [36] datasets. The GoPro dataset contains $2,103$ image pairs for training and $1,111$ image pairs for testing, while the HIDE dataset contains $2,025$ image pairs used only for testing. Additionally, we utilize the real-world RealBlur [31] dataset, which contains RealBlur-J and RealBlur-R subsets. Each subset contains $3,758$ training pairs and $980$ testing pairs, with RealBlur-J in JPEG and RealBlur-R in Raw format.

## 4.2 Experimental Results

**Quantitative Analysis.** As shown in Tab. 1, we compare the deblurring performance of four baselines and their BlurDM-enhanced versions, where "Baseline" and "BlurDM" refer to the deblurring performance without and with BlurDM, respectively. The results indicate that BlurDM consistently and significantly enhances deblurring performance, yielding average PSNR improvements of $0.31$ dB, $0.32$ dB, $0.78$ dB, and $0.69$ dB on the GoPro, HIDE, RealBlur-J, and RealBlur-R test sets, respectively. Additionally, BlurDM achieves average PSNR improvements of $0.59$ dB, $0.75$ dB, $0.25$ dB, and $0.51$ dB on MIMO-UNet, Stripformer, FFTformer, and LoFormer, respectively. Notably, BlurDM achieves substantial performance gains, up to $0.73$ dB, $1.16$ dB, $0.44$ dB, and $1.24$ dB for MIMO-UNet, Stripformer, FFTformer, and LoFormer, respectively. On average across all backbones and datasets, BlurDM achieves an overall gain of $0.53$ dB in PSNR, $0.004$ in SSIM, and a reduction of $0.0028$ in LPIPS. These comprehensive quantitative results demonstrate that BlurDM substantially enhances the performance of deblurring models across diverse datasets, highlighting BlurDM's effectiveness and robustness as a flexible prior generation network for image deblurring.

**Qualitative Analysis.** We provide qualitative comparisons of four baselines and their BlurDM-enhanced versions on the GoPro and HIDE test sets in Fig. 4 and RealBlur-J test set in Fig. 5. The results show that BlurDM consistently produces sharper and more visually appealing deblurred results than "Baseline." By integrating BlurDM into the latent space of a deblurring network, we leverage its ability to learn rich and realistic image priors while preserving the network's fidelity to sharp image contents.

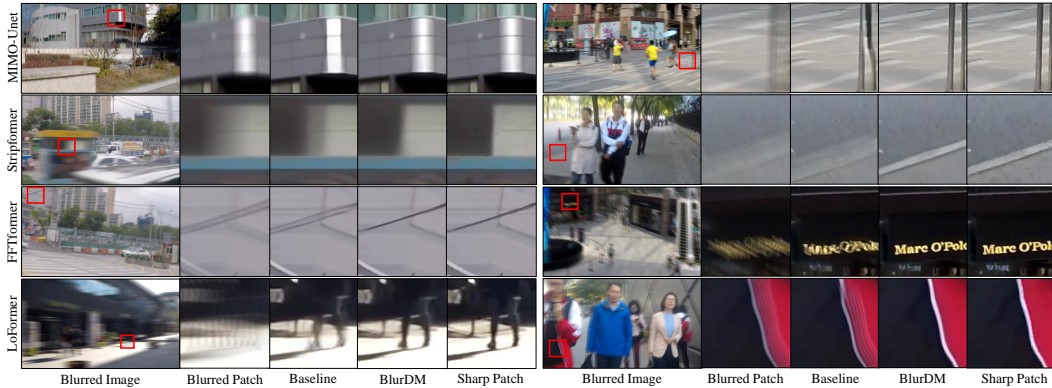

Figure 4: Qualitative results on the GoPro (left) and HIDE (right) datasets.

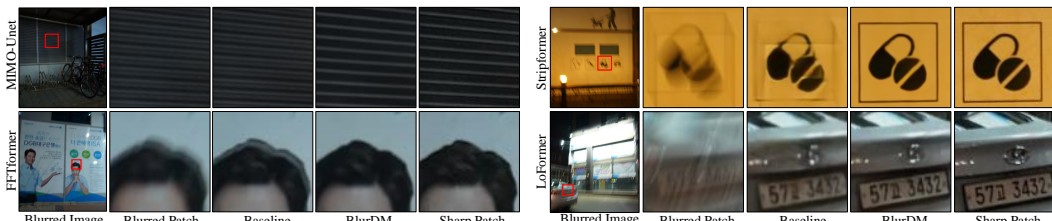

Figure 5: Qualitative results on the RealBlur-J dataset.

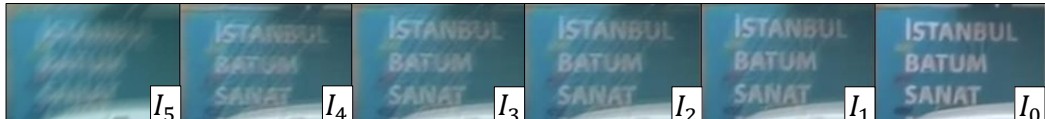

Figure 6: Deblurred results $I_5$ to $I_0$ from latent features $Z_5^B$ to $Z_0^B$, showing reduced blur as reverse steps increase.

## 4.3  Ablation studies

To evaluate the effect of the proposed components in BlurDM, we adopt MIMO-UNet as the baseline deblurring model and analyze its performance under various ablation settings. Specifically, we analyze the effectiveness of noise and blur residual estimators, compare different prior generation methods, analyze blur residual modeling in the latent space, inspect the effect of iteration counts, i.e., $T$, in BlurDM, compare different diffusion-based methods, analyze the effectiveness of each training stage, and measure the computational overhead introduced by BlurDM. All experiments are conducted with $1,000$ training epochs used.

**Effectiveness of Noise and Blur Estimators.**  We evaluate the effectiveness of the noise and blur estimators in BlurDM through an ablation study shown in Tab. 2. "Net1" denotes the baseline deblurring model. "Net2" represents a conventional DDPM-based design using only the noise estimator. "Net3" denotes a BlurDM variant that estimates blur residuals but omits the noise component. "Net4" is our complete BlurDM design incorporating both estimators. As can be seen, both "Net2" and "Net3" improve performance over the baseline, which demonstrates the individual benefit of noise and blur estimation. Moreover, "Net4" achieves the best result, showing that combining both estimators yields complementary gains. These findings further confirm the importance of explicitly modeling blur residuals to improve deblurring effectiveness.

**Comparison of Prior Generation Methods.**  We evaluate the performance of the baseline deblurring model enhanced by various prior generation methods, including MLP, DDPM [10], RDDM [17], and our proposed BlurDM, on the GoPro and RealBlur-J datasets (see Tab. 3). "Net1" denotes the baseline model without guidance by a prior generation network. "Net2" denotes the deblurring model enhanced with MLP layers, without using a diffusion process. "Net3", "Net4", and "Net5" corre-

Table 2: Effectiveness of the noise estimator and the blur estimator on the GoPro test set.

|      | Noise Estimator | Blur Estimator | PSNR |
|------|:---:|:---:|------|
| Net1 |   |   | 31.78 |
| Net2 | ✓ |   | 31.91 |
| Net3 |   | ✓ | 32.20 |
| Net4 | ✓ | ✓ | **32.28** |

Table 3: Comparison of different prior generators on GoPro and RealBlur-J datasets in PSNR.

|      | Prior Generators | GoPro | RealBlur-J |
|------|:---:|:---:|:---:|
| Net1 | N/A | 31.78 | 31.59 |
| Net2 | MLP | 31.90 | 31.84 |
| Net3 | DDPM | 31.91 | 31.85 |
| Net4 | RDDM | 32.03 | 31.90 |
| Net5 | BlurDM | **32.28** | **32.13** |

spond to the deblurring models enhanced by different diffusion-based priors, including DDPM [10], RDDM [17], and the proposed BlurDM, respectively.

While integrating the standard diffusion process (DDPM) into the deblurring model improves performance compared to the baseline ("Net3" vs. "Net1"), the gain is comparable to that of "Net2," which uses the same MLP structure without diffusion. This suggests that the standard diffusion process alone contributes little to deblurring performance. In contrast, BlurDM explicitly incorporates the blur formation process into diffusion, leading to superior performance over both the standard diffusion-based prior ("Net5" vs. "Net3") and the residual diffusion prior (RDDM), which lacks the proposed blur-aware diffusion mechanism ("Net5" vs. "Net4").

**Analysis of Blur Residual Modeling in Latent Space.** To verify whether BlurDM models blur formation in the latent space, we analyze outputs at different reverse diffusion steps during inference. While the model is trained with $T = 5$ steps, we evaluate intermediate latent representations by performing $t = [0, 1, 2, 3, 4, 5]$ reverse steps from the fully blurred latent $Z_5^B$, yielding a sequence $[Z_5^B, Z_4^B, \ldots, Z_0^B]$. Each $Z_t^B$ is decoded into an image $I_t$ via the deblurring network. As illustrated in Fig. 6, the outputs transition progressively from blurred ($I_5$) to sharp ($I_0$), confirming that BlurDM's latent representation captures a progressive blur-to-sharp structure and enables interpretable modeling in latent space.

**Effect of Iteration Counts in BlurDM.** Compared to the standard diffusion model in [10], which requires thousands of iterations in the reverse generation process, applying diffusion networks in the latent space has proven effective in reducing the number of iterations [4, 47]. Therefore, we examine the effect of different iteration counts used in BlurDM on deblurring performance, as shown in Fig. 7. Specifically, we test eight iteration settings $T \in \{0, 1, 2, 4, 5, 6, 8, 10\}$ and evaluate their deblurring performances on the GoPro test set. The results show that BlurDM significantly improves performance with two iterations and reaches peak performance after five, showcasing its ability to achieve substantial and stable performance gains with only a few iterations.

**Analysis of BlurDM's Computational Overhead.** We present the computational overhead introduced by BlurDM in Tab. 4, measuring FLOPs and inference time on a $256 \times 256$ image using an NVIDIA GeForce RTX 3090. The results show that BlurDM introduces only a slight increase in computational complexity while significantly improving deblurring performance. Specifically, BlurDM adds an average of just 4.16G FLOPs, 3.33M parameters, and 9 milliseconds across four deblurring models, demonstrating its effectiveness with minimal overhead. Note that the number of parameters varies across different deblurring models, as PFM must adapt to the varying channel dimensions of each decoder.

**Comparison of different diffusion-based methods.** We compare BlurDM with two recent diffusion-based approaches, HI-Diff [4] and RDDM [18], in Tab. 5. HI-Diff follows the conventional diffusion process in the latent space, where Gaussian noise is progressively injected directly into the latent until it becomes pure noise, and a learned reverse process is used to reconstruct a clean latent for deblurring. RDDM first forms an image space residual by subtracting the clean image from the degraded one, then performs diffusion on the clean image that jointly models this residual and Gaussian noise. Both HI-Diff and RDDM neglect the physics of blur formation. BlurDM addresses this gap by explicitly integrating the blur formation process with diffusion, executing dual noise and blur diffusion that matches the physics of blur accumulation. With comparable parameter counts and FLOPs, BlurDM consistently delivers higher PSNR and SSIM. As a plug-and-play module, it integrates seamlessly with diverse backbone architectures, demonstrating both strong performance and broad generalizability.

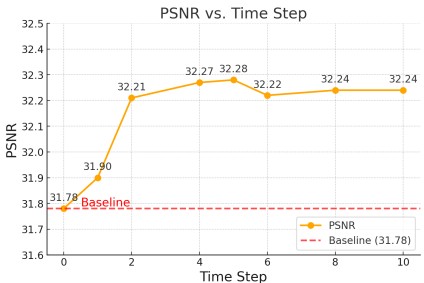

Figure 7: Effect of the number of iterations in BlurDM on deblurring performance in PSNR on the GoPro dataset.

Table 4: Computational overhead comparison between baseline deblurring models and their BlurDM-enhanced versions.

| Method | | FLOPs (G) | Params (M) | Time (ms) |
|---|---|---|---|---|
| MIMO-Unet+ | Baseline | 153.93 | 16.11 | 31 |
| | +BlurDM | 158.10 (+4.17) | 18.29 (+2.18) | 42 (+11) |
| Stripformer | Baseline | 170.02 | 19.71 | 48 |
| | +BlurDM | 174.18 (+4.16) | 24.33 (+4.62) | 55 (+7) |
| FFTformer | Baseline | 131.53 | 14.88 | 131 |
| | +BlurDM | 135.69 (+4.16) | 18.66 (+3.78) | 141 (+10) |
| LoFormer-S | Baseline | 52.19 | 16.35 | 93 |
| | +BlurDM | 56.35 (+4.16) | 19.08 (+2.73) | 99 (+6) |

Table 5: Comparison of different diffusion-based methods on the GoPro dataset.

| Method | PSNR | SSIM | Params (M) | FLOPs (G) |
|---|---|---|---|---|
| HI-Diff | 33.33 | 0.955 | 23.99 | 125.47 |
| RDDM | 32.40 | 0.963 | 15.49 | 134.20 |
| BlurDM (Stripformer) | 33.53 | 0.966 | 24.33 | 174.18 |
| BlurDM (FFTformer) | 34.34 | 0.970 | 18.66 | 135.69 |
| BlurDM (LoFormer) | 33.70 | 0.967 | 19.08 | 56.35 |

Table 6: Effect of each training stage on the GoPro dataset.

| Model | Stage 1 | Stage 2 | Stage 3 | PSNR |
|---|---|---|---|---|
| Net1 | | | | 31.78 (baseline) |
| Net2 | ✓ | | | 32.69 (upper bound) |
| Net3 | | | ✓ | 31.80 |
| Net4 | ✓ | | ✓ | 32.01 |
| Net5 | ✓ | ✓ | | 31.95 |
| Net6 | ✓ | ✓ | ✓ | **32.28** |

**Effectiveness of Each Training Stage.** We evaluate the effectiveness of the three-stage training strategy, as shown in Tab. 6. "Net1" denotes the baseline deblurring performance without incorporating BlurDM. In "Net2", the ground-truth sharp image is passed through the Sharp Encoder to obtain the sharp prior $Z^S$, which is then used by BlurDM. This setting serves as an upper bound on achievable deblurring performance with an ideal prior. In "Net3", we jointly optimize BlurDM and the deblurring model without pretraining through Stage 1 and Stage 2, serving as a baseline for a purely data-driven approach. In "Net4" and "Net5", after completing pre-training in Stage 1, we apply either Stage 2 or Stage 3 alone to optimize BlurDM and the deblurring model. "Net6" employs the full three-stage training pipeline, achieving the highest PSNR among all settings. These results clearly demonstrate the effectiveness and necessity of the proposed three-stage training strategy in improving deblurring performance.

## 5 Limitations

Since BlurDM is designed based on the motion blur formation process, it effectively handles blur caused by camera motion and moving objects. However, it may not be well-suited for handling defocus blur, which arises from optical aberrations due to out-of-focus issues. Unlike motion blur, defocus blur is depth-dependent and does not exhibit the same temporal accumulation properties, making it fundamentally different in nature. Addressing defocus deblurring would require a distinct approach, potentially incorporating depth estimation or optical defocus modeling, which remains an open direction for future research.

## 6 Conclusion

We proposed Blur Diffusion Model (BlurDM), a novel diffusion-based framework for image deblurring. BlurDM integrates the blur formation process into the diffusion framework, simultaneously performing noise diffusion and blur diffusion for more effective deblurring. In the forward process, BlurDM progressively degrades a sharp image by introducing both noise and blur through a dual noise and blur diffusion process. Conversely, in the reverse process, BlurDM restores the image by removing noise and blur residuals via its dual denoising and deblurring process. To enhance the performance of existing deblurring networks, we incorporated BlurDM into their latent spaces as a prior generator, seamlessly integrating the learned prior into each decoder block via our proposed Prior Fusion Module (PFM) to generate higher-quality deblurring results. Extensive experimental results have demonstrated that our method effectively improves deblurring performance across four deblurring models on four deblurring datasets.

# 7 Acknowledgments

This work was supported in part by the National Science and Technology Council (NSTC) under grants 114-2221-E-A49-038-MY3, 112-2221-E-A49-090-MY3, 113-2221-E-004-001-MY3, 113-2634-F-002-008, and 114-2221-E-007-065-MY3, and by the NVIDIA Taiwan AI Research & Development Center (TRDC).

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

# A   Appendices

## A.1   One-Step Diffusion Derivation for BlurDM

In the dual noise and blur diffusion process of BlurDM, the forward process is defined as:

$$q(I_{1:T} \mid I_0, e_{1:T}) := \prod_{t=1}^{T} q(I_t \mid I_{t-1}, e_t);$$

$$q(I_t \mid I_{t-1}, e_t) := \mathcal{N}\left(I_t;\ \frac{\alpha_{t-1}}{\alpha_t} I_{t-1} + \frac{1}{\alpha_t} e_t,\ \beta_t^2 \mathbf{I}\right), \tag{13}$$

where $e_t = \int_{\tau=\alpha_{t-1}}^{\alpha_t} H(\tau)\, d\tau$ is the blur residual accumulated during the exposure interval $[\alpha_{t-1}, \alpha_t]$.

We now expand the full forward process by recursively substituting the previous states. Starting from the last time step as

$$
\begin{aligned}
I_T &= \frac{\alpha_{T-1}}{\alpha_T} I_{T-1} + \frac{1}{\alpha_T} e_T + \beta_T \epsilon_T \\
&= \frac{\alpha_{T-1}}{\alpha_T}\left(\frac{\alpha_{T-2}}{\alpha_{T-1}} I_{T-2} + \frac{1}{\alpha_{T-1}} e_{T-1} + \beta_{T-1}\epsilon_{T-1}\right) + \frac{1}{\alpha_T} e_T + \beta_T \epsilon_T \\
&= \frac{\alpha_{T-2}}{\alpha_T} I_{T-2} + \frac{1}{\alpha_T}(e_{T-1} + e_T) + \frac{\alpha_{T-1}}{\alpha_T}\beta_{T-1}\epsilon_{T-1} + \beta_T \epsilon_T \\
&\ \ \vdots \\
&= \frac{\alpha_0}{\alpha_T} I_0 + \sum_{t=1}^{T} \frac{1}{\alpha_T} e_t + \sum_{t=1}^{T} \frac{\alpha_t}{\alpha_T} \beta_t \epsilon_t.
\end{aligned}
\tag{14}
$$

Since $\epsilon_t \sim \mathcal{N}(0, \mathbf{I})$ are independent for all $t$, their weighted sum remains Gaussian with zero mean. The resulting variance of this sum is

$$\bar{\beta}_T^2 = \sum_{t=1}^{T}\left(\frac{\alpha_t}{\alpha_T}\right)^2 \beta_t^2,$$

which allows us to reparameterize as:

$$\sum_{t=1}^{T} \frac{\alpha_t}{\alpha_T} \beta_t \epsilon_t = \bar{\beta}_T \epsilon, \quad \epsilon \sim \mathcal{N}(0, \mathbf{I}).$$

Moreover, combining the clean image component and the accumulated blur residuals, we observe that

$$\frac{\alpha_0}{\alpha_T} I_0 + \frac{1}{\alpha_T} \sum_{t=1}^{T} e_t = \frac{\alpha_0}{\alpha_T} \cdot \frac{1}{\alpha_0} \int_0^{\alpha_0} H(\tau)\, d\tau + \frac{1}{\alpha_T} \int_{\alpha_0}^{\alpha_T} H(\tau)\, d\tau = \frac{1}{\alpha_T} \int_0^{\alpha_T} H(\tau)\, d\tau = B,$$

where $B$ denotes the fully blurred image formed by integrating the instantaneous scene radiance $H(\tau)$ over the total exposure time interval $[0, \alpha_T]$.

Thus, the forward process simplifies to a single-step form:

$$I_T = B + \bar{\beta}_T \epsilon, \tag{15}$$

with the corresponding marginal distribution

$$q(I_T \mid I_0, e_{1:T}) = \mathcal{N}\left(I_T;\ \frac{1}{\alpha_T} I_0 + \frac{1}{\alpha_T} \sum_{t=1}^{T} e_t,\ \bar{\beta}_T^2 \mathbf{I}\right). \tag{16}$$

This one-step form is mathematically equivalent to the full forward process, while providing a more computationally efficient approximation that captures both accumulated blur and noise in a single Gaussian transition.

## A.2 ELBO and Optimization for BlurDM

To reconstruct the sharp image $I_0$ from the degraded observation $I_T$, we adopt the variational inference framework of DDPM [10] and derive an evidence lower bound (ELBO) that explicitly incorporates the blur residuals $e_{1:T}$. The joint ELBO is expressed as

$$\log p_\theta(I_0) \geq \mathbb{E}_{q(I_{1:T}|I_0,e_{1:T})}\left[\log \frac{p_\theta(I_{0:T})}{q(I_{1:T}|I_0,e_{1:T})}\right] =: \mathcal{L}_{\text{ELBO}}. \tag{17}$$

By unrolling the Markov chain formulation in DDPM, we can rewrite the objective as:

$$\mathcal{L}_{\text{ELBO}} = \mathbb{E}_q\left[-\log \frac{p_\theta(I_{0:T})}{q(I_{1:T} \mid I_0, e_{1:T})}\right] \tag{18}$$

$$= \mathbb{E}_q\left[-\log p_\theta(I_T) -\sum_{t\geq 1}\log \frac{p_\theta(I_{t-1} \mid I_t)}{q(I_t \mid I_{t-1}, e_t)}\right] \tag{19}$$

$$= \mathbb{E}_q\left[-\log p_\theta(I_T) -\sum_{t>1}\log \frac{p_\theta(I_{t-1} \mid I_t)}{q(I_t \mid I_{t-1}, e_t)} - \log \frac{p_\theta(I_0 \mid I_1)}{q(I_1 \mid I_0, e_1)}\right] \tag{20}$$

$$= \mathbb{E}_q\left[-\log p_\theta(I_T) -\sum_{t>1}\log \frac{p_\theta(I_{t-1} \mid I_t)}{q(I_{t-1} \mid I_t, I_0, e_{1:t})}\frac{q(I_{t-1} \mid I_0, e_{1:t})}{q(I_t \mid I_0, e_{1:t})} - \log \frac{p_\theta(I_0 \mid I_1)}{q(I_1 \mid I_0, e_1)}\right] \tag{21}$$

$$= \mathbb{E}_q\left[-\log \frac{p_\theta(I_T)}{q(I_T \mid I_0, e_{1:T})} -\sum_{t>1}\log \frac{p_\theta(I_{t-1} \mid I_t)}{q(I_{t-1} \mid I_t, I_0, e_{1:t})} - \log p_\theta(I_0 \mid I_1)\right]. \tag{22}$$

Rewriting (22) in terms of KL-divergence yields:

$$\mathcal{L}_{\text{ELBO}} \tag{23}$$

$$= \mathbb{E}_q\left[D_{\text{KL}}(q(I_T \mid I_0, e_{1:T}) \parallel p_\theta(I_T)) + \sum_{t\geq 1}D_{\text{KL}}(q(I_{t-1} \mid I_t, I_0, e_{1:t}) \parallel p_\theta(I_{t-1} \mid I_t)) - \log p_\theta(I_0 \mid I_1)\right]. \tag{24}$$

Unlike standard diffusion models, which assume a standard Gaussian prior at the terminal state, our model defines the prior distribution as:

$$p_\theta(I_T) = \mathcal{N}(I_T; \; B, \; \bar{\beta}_T^2 \mathbf{I}),$$

where $B$ denotes the physically blurred image obtained from full exposure integration. This formulation ensures that the terminal distribution is aligned with the mean of the forward marginal $q(I_T \mid I_0, e_{1:T})$. Therefore, our prior is not an arbitrary isotropic Gaussian but a noise-perturbed version of a blurry observation, consistent with the forward process structure.

Given this structural alignment between the forward marginal and the prior, we follow the standard approach of DDPM [10] and DDIM [38], and omit the terminal KL divergence $D_{\text{KL}}(q(I_T \mid I_0, e_{1:T}) \parallel p_\theta(I_T))$ during training. Instead, we retain only the stepwise KL divergence terms, which capture the discrepancy between the reverse and forward transitions at each timestep as:

$$\sum_{t\geq 1}D_{\text{KL}}(q(I_{t-1} \mid I_t, I_0, e_{1:t}) \parallel p_\theta(I_{t-1} \mid I_t)). \tag{25}$$

To compute these terms, we derive the posterior $q(I_{t-1} \mid I_t, I_0, e_{1:t})$ using Bayes' rule:

$$q(I_{t-1} \mid I_t, I_0, e_{1:t}) = q(I_t \mid I_{t-1}, I_0, e_{1:t}) \cdot \frac{q(I_{t-1} \mid I_0, e_{1:t})}{q(I_t \mid I_0, e_{1:t})}. \tag{26}$$

From (6), we have

$$q(I_{t-1} \mid I_0, e_{1:t}) = \mathcal{N}\left(I_{t-1}; \frac{1}{\alpha_{t-1}}I_0 + \frac{1}{\alpha_{t-1}}\sum_{i=1}^{t-1}e_i, \; \bar{\beta}_{t-1}^2 \mathbf{I}\right).$$

From (5), the forward transition is expressed as

$$q(I_t \mid I_{t-1}, e_t) = \mathcal{N}\left(I_t; \frac{\alpha_{t-1}}{\alpha_t}I_{t-1} + \frac{1}{\alpha_t}e_t, \bar{\beta}_t^2\mathbf{I}\right).$$

Combining the above, we obtain the posterior

$$q(I_{t-1} \mid I_t, I_0, e_{1:t}) = \mathcal{N}\left(I_{t-1}; \mu_t(I_t, I_0, e_{1:t}), \sigma_t^2(I_t, I_0, e_{1:t})\mathbf{I}\right), \tag{27}$$

$$\propto \exp\left(-\frac{1}{2}\left(\frac{(I_t - \frac{\alpha_{t-1}}{\alpha_t}I_{t-1} - \frac{1}{\alpha_t}e_t)^2}{\beta_t^2} + \frac{(I_{t-1} - \frac{1}{\alpha_{t-1}}I_0 - \frac{1}{\alpha_{t-1}}\sum_{i=1}^{t-1}e_i)^2}{\bar{\beta}_{t-1}^2} - \frac{(I_t - \frac{1}{\alpha_t}I_0 - \frac{1}{\alpha_t}\sum_{i=1}^{t}e_i)^2}{\bar{\beta}_t^2}\right)\right), \tag{28}$$

which can be further simplified to

$$\propto \exp\left(-\frac{1}{2}\left(\frac{\bar{\beta}_t^2}{\beta_t^2\bar{\beta}_{t-1}^2}I_{t-1}^2 - 2\left(\frac{\alpha_{t-1}}{\alpha_t\beta_t^2}I_t - \frac{\alpha_{t-1}}{\alpha_t^2\beta_t^2}e_t + \frac{1}{\alpha_{t-1}\bar{\beta}_{t-1}^2}I_0 + \frac{1}{\alpha_{t-1}\bar{\beta}_{t-1}^2}\sum_{i=1}^{t-1}e_i\right)I_{t-1} + C\right)\right), \tag{29}$$

where $C = C(I_t, I_0, e_{1:t})$ denotes the terms not involving $I_{t-1}$.

From (29), the posterior parameters are given by

$$\mu_t(I_t, I_0, e_{1:t}) = \frac{\frac{\alpha_{t-1}}{\alpha_t\beta_t^2}I_t - \frac{\alpha_{t-1}}{\alpha_t^2\beta_t^2}e_t + \frac{1}{\alpha_{t-1}\bar{\beta}_{t-1}^2}I_0 + \frac{1}{\alpha_{t-1}\bar{\beta}_{t-1}^2}\sum_{i=1}^{t-1}e_i}{\frac{\bar{\beta}_t^2}{\beta_t^2\bar{\beta}_{t-1}^2}} \tag{30}$$

$$= \frac{\alpha_t}{\alpha_{t-1}}I_t - \frac{1}{\alpha_t}e_t - \frac{\alpha_t}{\alpha_{t-1}}\cdot\frac{\beta_t^2}{\bar{\beta}_t}\epsilon, \tag{31}$$

$$\sigma_t^2(I_t, I_0, e_{1:t}) = \frac{\beta_t^2\bar{\beta}_{t-1}^2}{\bar{\beta}_t^2}, \tag{32}$$

where $\mu_t(I_t, I_0, e_{1:t})$ denotes the posterior mean, which is derived by combining (6) and (4) through the product of Gaussian densities. $\sigma_t^2(I_t, I_0, e_{1:t})$ represents the corresponding posterior variance.

We model the reverse process beginning at

$$p_\theta(I_T) = \mathcal{N}(I_T; B, \bar{\beta}_T^2\mathbf{I}),$$

and define

$$p_\theta(I_{t-1} \mid I_t) = q(I_{t-1} \mid I_t, I_0^\theta, I_{\text{res}}^\theta).$$

In our setting, since the variances of the two Gaussian distributions are matched exactly, the KL divergence reduces to a squared difference between their means, as is standard in DDPM [10]. Accordingly, the KL divergence term in (25) reduces to

$$D_{\text{KL}}(q(I_{t-1} \mid I_t, I_0, e_{1:t}) \parallel p_\theta(I_{t-1} \mid I_t)) = \mathbb{E}\left[\left\|\mu_t - \mu_t^\theta\right\|^2\right], \tag{33}$$

where the mean of the true posterior is given by

$$\mu_t = \frac{\alpha_t}{\alpha_{t-1}}I_t - \frac{1}{\alpha_t}e_t - \frac{\alpha_t}{\alpha_{t-1}}\frac{\beta_t^2}{\bar{\beta}_t}\epsilon,$$

and the model-predicted mean is

$$\mu_t^\theta = \frac{\alpha_t}{\alpha_{t-1}}I_t - \frac{1}{\alpha_t}e^\theta(I_t, t, B) - \frac{\alpha_t}{\alpha_{t-1}}\frac{\beta_t^2}{\bar{\beta}_t}\epsilon^\theta(I_t, t, B),$$

where $e^\theta(I_t, t, B)$ and $\epsilon^\theta(I_t, t, B)$ denote the learned blur residual and noise estimators, respectively.

Based on (4) and (33), we can derive the following optimization objectives

$$L_{e_t}(\theta) = \mathbb{E}\left[\lambda_e\left\|e_t - e_t^\theta(I_t, t, B)\right\|^2\right], \tag{34}$$

$$L_\epsilon(\theta) = \mathbb{E}\left[\lambda_\epsilon\left\|\epsilon - \epsilon^\theta(I_t, t, B)\right\|^2\right]. \tag{35}$$

Thus, the optimization of BlurDM reduces to minimizing the combined loss in (34) and (35), which directly supervises both the blur residual estimator and the noise residual estimator through their respective ground-truth signals.

**End-to-End Trajectory Supervision via Final Reconstruction**  Although per-step ground-truth blur residuals are unavailable, recent diffusion-based research [4, 8, 35, 47] has shown that supervising the generated results is sufficient to train the diffusion model.

Following this line of evidence, we train BlurDM with the reconstruction objective

$$\mathcal{L}_{\text{rec}} = \left\| I_0^\theta - I_0 \right\|, \tag{36}$$

where $I_0^\theta$ is obtained by successively denoising the degraded observation $I_T$ through $T$ learned reverse steps.

At each step $t \in \{T, \ldots, 1\}$, the network predicts a blur residual $\hat{e}_t^\theta = e^\theta(I_t, t, B)$ and a noise residual $\hat{\epsilon}_t^\theta = \epsilon^\theta(I_t, t, B)$, then reconstructs the previous latent state via

$$I_{t-1}^\theta = \frac{\alpha_t}{\alpha_{t-1}} I_t - \frac{1}{\alpha_{t-1}} \hat{e}_t^\theta - \frac{\alpha_t}{\alpha_{t-1}} \frac{\beta_t^2}{\bar{\beta}_t} \hat{\epsilon}_t^\theta. \tag{37}$$

This yields the unrolled trajectory

$$I_T^\theta = I_T, \tag{38}$$
$$I_{t-1}^\theta = g_t^\theta(I_t^\theta), \qquad t = T, \ldots, 1, \tag{39}$$
$$I_0^\theta = g_1^\theta \circ \cdots \circ g_T^\theta(I_T), \tag{40}$$

where every step operator $g_t^\theta$ shares parameters $\theta$. Backpropagating $\mathcal{L}_{\text{rec}}$ supplies gradients to *all* intermediate residual predictions $\{\hat{e}_t^\theta, \hat{\epsilon}_t^\theta\}_{t=1}^T$, allowing amortized trajectory level optimization despite the lack of explicit stepwise supervision. This strategy mirrors the unrolled inference paradigm in generative models based on variation and scores and empirically produces stable convergence with strong deblurring fidelity.

## A.3 Deterministic Implicit Sampling for BlurDM

In this section, we provide a formal derivation to demonstrate that our deterministic reverse process

$$q_\sigma(I_{t-1} \mid I_t, I_0, e_{1:t}),$$

preserves the forward process distribution defined in (6), i.e.,

$$q(I_t \mid I_0, e_{1:t}) = \mathcal{N}\left(I_t; \frac{\alpha_0}{\alpha_t} I_0 + \frac{1}{\alpha_t} \sum_{i=1}^t e_i, \bar{\beta}_t^2 \mathbf{I}\right).$$

We follow the approach of DDIM [38] and proceed by mathematical induction from $t = T$ to $t = 1$. Assuming that the marginal distribution $q(I_t \mid I_0, e_{1:t})$ is valid at step $t$, we aim to prove that sampling $I_{t-1}$ from $q_\sigma(I_{t-1} \mid I_t, I_0^\theta, e_{1:t})$ yields a distribution consistent with

$$q(I_{t-1} \mid I_0, e_{1:t-1}) = \mathcal{N}\left(I_{t-1}; \frac{\alpha_0}{\alpha_{t-1}} I_0 + \frac{1}{\alpha_{t-1}} \sum_{i=1}^{t-1} e_i, \bar{\beta}_{t-1}^2 \mathbf{I}\right).$$

Let us begin by rewriting the marginal at time $t$

$$q(I_t \mid I_0, e_{1:t}) = \mathcal{N}\left(I_t; \frac{\alpha_0}{\alpha_t} I_0 + \frac{1}{\alpha_t} \sum_{i=1}^t e_i, \bar{\beta}_t^2 \mathbf{I}\right). \tag{41}$$

We define the reverse transition distribution using the deterministic implicit sampling formulation:

$$q_\sigma(I_{t-1} \mid I_t, I_0, e_{1:t}) = \tag{42}$$

$$\mathcal{N}\left(I_{t-1}; \underbrace{\frac{\alpha_0}{\alpha_{t-1}} I_0 + \frac{1}{\alpha_{t-1}} \sum_{i=1}^{t-1} e_i + \sqrt{\bar{\beta}_{t-1}^2 - \sigma_t^2} \cdot \frac{I_t - \left(\frac{\alpha_0}{\alpha_t} I_0^\theta + \frac{1}{\alpha_t} \sum_{i=1}^t e_i\right)}{\bar{\beta}_t}}_{\mu_{t-1}}, \sigma_t^2 \mathbf{I}\right). \tag{43}$$

We now compute the implied marginal distribution over $I_{t-1}$ by integrating out $I_t$, using the properties of marginal and conditional Gaussians. Let the conditional be $p(I_{t-1} \mid I_t)$ and the marginal $q(I_t)$, then the marginal of $I_{t-1}$ becomes

$$q(I_{t-1} \mid I_0, e_{1:t}) = \int q_\sigma(I_{t-1} \mid I_t, I_0, e_{1:t}) \cdot q(I_t \mid I_0, e_{1:t}) \, dI_t.$$

By applying the formula for Gaussian marginalization over linear Gaussian transformations, we obtain

**Mean:**

$$\mu_{t-1} = \frac{\alpha_0}{\alpha_{t-1}} I_0 + \frac{1}{\alpha_{t-1}} \sum_{i=1}^{t-1} e_i + \sqrt{\bar{\beta}_{t-1}^2 - \sigma_t^2} \cdot \frac{\left(\frac{\alpha_0}{\alpha_t} I_0 + \frac{1}{\alpha_t} \sum_{i=1}^{t} e_i\right) - \left(\frac{\alpha_0}{\alpha_t} I_0 + \frac{1}{\alpha_t} \sum_{i=1}^{t} e_i\right)}{\bar{\beta}_t} \tag{44}$$

$$= \frac{\alpha_0}{\alpha_{t-1}} I_0 + \frac{1}{\alpha_{t-1}} \sum_{i=1}^{t-1} e_i, \tag{45}$$

with variance term $\sigma_t^2 = \eta \cdot \frac{\beta_t^2 \bar{\beta}_{t-1}^2}{\bar{\beta}_t^2}$. When $\eta = 0$, this yields a deterministic sampling.

**Variance:**

$$\sigma_{t-1}^2 \mathbf{I} = \sigma_t^2 \mathbf{I} + \left(\frac{\sqrt{\bar{\beta}_{t-1}^2 - \sigma_t^2}}{\bar{\beta}_t}\right)^2 \bar{\beta}_t^2 \mathbf{I} \tag{46}$$

$$= \sigma_t^2 \mathbf{I} + \left(\frac{\bar{\beta}_{t-1}^2 - \sigma_t^2}{\bar{\beta}_t^2}\right) \bar{\beta}_t^2 \mathbf{I} \tag{47}$$

$$= \bar{\beta}_{t-1}^2 \mathbf{I}. \tag{48}$$

Hence, the marginal distribution at step $t-1$ becomes

$$q(I_{t-1} \mid I_0, e_{1:t-1}) = \mathcal{N}\left(I_{t-1}; \frac{\alpha_0}{\alpha_{t-1}} I_0 + \frac{1}{\alpha_{t-1}} \sum_{i=1}^{t-1} e_i, \bar{\beta}_{t-1}^2 \mathbf{I}\right),$$

which confirms that (6) holds at step $t-1$.

By induction, the deterministic sampling formulation maintains consistency with the original forward process distribution at every timestep.

**Derivation from (9) to (10)**   Based on (9), we can sample $I_{t-1}$ from $q_\sigma(I_{t-1} \mid I_t, I_0^\theta, e_{1:t}^\theta)$ as

$$I_{t-1} = \frac{\alpha_0}{\alpha_{t-1}} I_0^\theta + \frac{1}{\alpha_{t-1}} \sum_{i=1}^{t-1} e_i^\theta + \sqrt{\bar{\beta}_{t-1}^2 - \sigma_t^2} \frac{I_t - (\frac{\alpha_0}{\alpha_t} I_0^\theta + \frac{1}{\alpha_t} \sum_{i=1}^{t} e_i^\theta)}{\bar{\beta}_t} + \sigma_t, \tag{49}$$

where $\sigma_t^2 = \eta \cdot \frac{\beta_t^2 \bar{\beta}_{t-1}^2}{\bar{\beta}_t^2}$, we set $\eta = 0$ for the deterministic sampling. In addition, we substitue $I_0^\theta$ with (8). Therefore, (49) can be rewritten as

$$I_{t-1} = \frac{\alpha_0}{\alpha_{t-1}} \left(\frac{\alpha_t}{\alpha_0} I_t - \frac{1}{\alpha_0} \sum_{i=1}^{t} e_i^\theta - \frac{\alpha_t}{\alpha_0} \bar{\beta}_t \epsilon^\theta\right) + \frac{1}{\alpha_{t-1}} \sum_{i=1}^{t-1} e_i^\theta \tag{50}$$

$$+ \bar{\beta}_{t-1} \frac{I_t - (\frac{\alpha_0}{\alpha_t}(\frac{\alpha_t}{\alpha_0} I_t - \frac{1}{\alpha_0} \sum_{i=1}^{t} e_i^\theta - \frac{\alpha_T}{\alpha_0} \bar{\beta}_t \epsilon^\theta) + \frac{1}{\alpha_t} \sum_{i=1}^{t} e_i^\theta)}{\bar{\beta}_t}, \tag{51}$$

$$= \frac{\alpha_t}{\alpha_{t-1}} I_t - \frac{1}{\alpha_{t-1}} e^\theta(I_t, t, B) - \left(\frac{\alpha_t \bar{\beta}_t}{\alpha_{t-1}} - \bar{\beta}_{t-1}\right) \epsilon^\theta(I_t, t, B). \tag{52}$$

Table 7: Comparison of training cost and performance between baseline and BlurDM.

| Method | Training epoch | Training time [h] | PSNR [dB] |
|---|---|---|---|
| Baseline 1 | 3000 | 66.7 | 32.44 |
| Baseline 2 | 6000 | 133.4 | 32.51 |
| BlurDM 1 | 1500 (Stage 1) + 500 (Stage 2) + 1500 (Stage 3) | 70.7 | 32.62 |
| BlurDM 2 | 3000 (Stage 1) + 500 (Stage 2) + 1500 (Stage 3) | 104.1 | 32.71 |
| BlurDM 3 | 3000 (Stage 1) + 500 (Stage 2) + 3000 (Stage 3) | 141.4 | 32.93 |

## A.4 Theoretical justification of latent BlurDM.

Let $z_t = E(I_t)$ be the latent feature produced by an encoder $E(\cdot)$ from the blurred image $I_t$ at exposure step $t$. A first–order Taylor expansion of $E$ around $I_{t-1}$ gives

$$z_t \approx z_{t-1} + \mathbf{J}_E(I_{t-1})\,(I_t - I_{t-1}), \tag{53}$$

where $\mathbf{J}_E(I_{t-1})$ is the Jacobian of the encoder at $I_{t-1}$. From the image–space exposure model,

$$I_t - I_{t-1} = \left(\frac{\alpha_{t-1}}{\alpha_t} - 1\right)I_{t-1} \;+\; \frac{1}{\alpha_t}\,e_t \;+\; \beta_t\,\varepsilon_t, \tag{54}$$

with $e_t$ the blur residual and $\varepsilon_t$ the stochastic noise at step $t$. Substituting yields

$$z_t \approx z_{t-1} + \mathbf{J}_E(I_{t-1})\left[\left(\frac{\alpha_{t-1}}{\alpha_t} - 1\right)I_{t-1} + \frac{1}{\alpha_t}e_t + \beta_t\varepsilon_t\right]. \tag{55}$$

When exposure increments are small so that $\frac{\alpha_{t-1}}{\alpha_t} \approx 1$, the term proportional to $I_{t-1}$ vanishes and we obtain the latent–space dynamics

$$z_t \approx \left(\frac{\alpha_{t-1}}{\alpha_t}\right)z_{t-1} \;+\; \frac{1}{\alpha_t}\,e_t^\theta \;+\; \beta_t\,\varepsilon_t^\theta, \qquad e_t^\theta := \mathbf{J}_E(I_{t-1})\,e_t,\; \varepsilon_t^\theta := \mathbf{J}_E(I_{t-1})\,\varepsilon_t, \tag{56}$$

where $\theta$ denotes the learnable parameters of $E$. Thus, blur residuals and stochastic noise in image space are projected into the latent space with the same coefficients, justifying that the two estimators in BlurDM can learn these terms directly in the latent domain.

## A.5 Training cost of the three-stage training strategy

In Tab. 7, we report the training complexity introduced by BlurDM in terms of training time, using MIMO-UNet as the backbone on GoPro dataset. The default number of training epochs for MIMO-UNet is 3, 000, denoted as "Baseline 1". Increasing the training epochs to 6, 000, denoted as "Baseline 2", results in only a marginal improvement of 0.07 dB in PSNR. For BlurDM, we experiment with different training configurations, including using 1, 500 or 3, 000 epochs for Stage 1 and Stage 3, with 500 epochs for Stage 2. Both "BlurDM 1" and "BlurDM 2" outperform Baseline 2 while requiring less training time, demonstrating the effectiveness of the proposed three-stage training strategy. Finally, we adopt "BlurDM 3" as our final model, which increases training time by only 8% while achieving a 0.42 dB improvement in PSNR compared to "Baseline 2".

## A.6 Performance comparison of the number of steps used in RDDM and BlurDM

Tab. 8 presents a comparison of RDDM and BlurDM under varying numbers of diffusion steps, where PSNR and LPIPS are used to assess performance. RDDM achieves its peak PSNR of 32.08 dB at step 4 and the best LPIPS score of 0.0121 at step 10. In comparison, BlurDM achieves a higher peak PSNR of 32.28 dB at step 5 and a lower LPIPS score of 0.0113 at step 10. These results demonstrate that BlurDM consistently outperforms RDDM in terms of both distortion-based (PSNR) and perceptual (LPIPS) quality metrics.

## A.7 Visualizations of Blur Residual

In Fig. 9, we compare blurred images at different exposure times synthesized from the GoPro [24] dataset and the corresponding blur residuals obtained using BlurDM. Here, we depict $T = 0$ as sharp images. As $T$ increases (i.e., as the exposure time increases), the images gradually become more blurred, and the corresponding blur residuals evolve accordingly. Eventually, at $T = 1$, the images correspond to the fully blurred versions.

Table 8: Performance under different step counts on the GoPro dataset.

| Method | 2 | 4 | 5 | 6 | 8 | 10 |
|---|---|---|---|---|---|---|
| | PSNR↑ / LPIPS↓ | PSNR↑ / LPIPS↓ | PSNR↑ / LPIPS↓ | PSNR↑ / LPIPS↓ | PSNR↑ / LPIPS↓ | PSNR↑ / LPIPS↓ |
| RDDM | 32.07 / 0.0130 | 32.08 / 0.0131 | 32.03 / 0.0125 | 32.01 / 0.0122 | 32.02 / 0.0122 | 32.02 / 0.0121 |
| BlurDM | **32.21 / 0.0116** | **32.27 / 0.0114** | **32.28 / 0.0114** | **32.22 / 0.0115** | **32.24 / 0.0113** | **32.24 / 0.0113** |

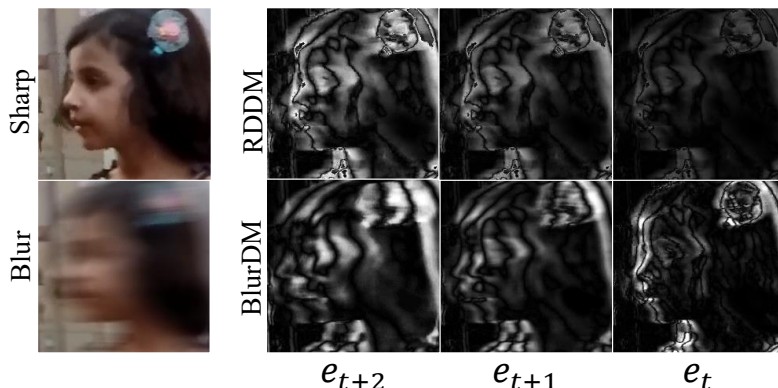

Figure 8: Comparison of blur residuals between RDDM and BlurDM at $t = 1$.

## A.8 Comparison between BlurDM and RDDM in Blur Residuals.

We visualize and compare blur residuals generated by BlurDM and RDDM [18] in Fig. 8. RDDM computes blur residuals using a simple subtraction operation followed by linear scaling to adjust intensity, resulting in residuals that primarily capture direct differences between blurred and sharp images, progressively magnified over time steps ($e_t$ to $e_{t+2}$). In contrast, BlurDM addresses the inductive bias of blur formation process to estimate blur residuals, capturing the non-linear progressive accumulation of blur. Unlike RDDM, BlurDM models how blurred residuals spatially diffuse and evolve over time, simulating the blur spread as exposure time extends. As a result, BlurDM more accurately represents the physical characteristics of motion blur within the diffusion process, leading to significantly improved deblurring performance.

## A.9 Deblurred Results on Real-world Datasets

We provide additional deblurred results for deblurring models using BlurDM, compared to those without using our method, referred to as Baseline. These deblurring models are trained on the GoPro [24] and RealBlur-J [31] training sets, and tested on the RealBlur-J testing set. We demonstrate qualitative comparisons based on four image deblurring models, including MIMO-UNet [5] in Fig. 10, Stripformer [40] in Fig. 11, FFTformer [13] in Fig. 12, and LoFormer [21] in Fig. 13.

## A.10 Broader Impacts

Our work improves the capability of image deblurring by introducing a diffusion-based framework that mimics the physical formation of motion blur. This has potential benefits in applications such as autonomous driving, medical imaging, and restoration of historical media. However, as with many image enhancement technologies, there exists a risk of misuse, such as reconstructing intentionally blurred faces or sensitive content, which may raise privacy concerns. We encourage the responsible and ethical use of deblurring models and suggest deploying them in contexts with appropriate privacy safeguards and user consent.

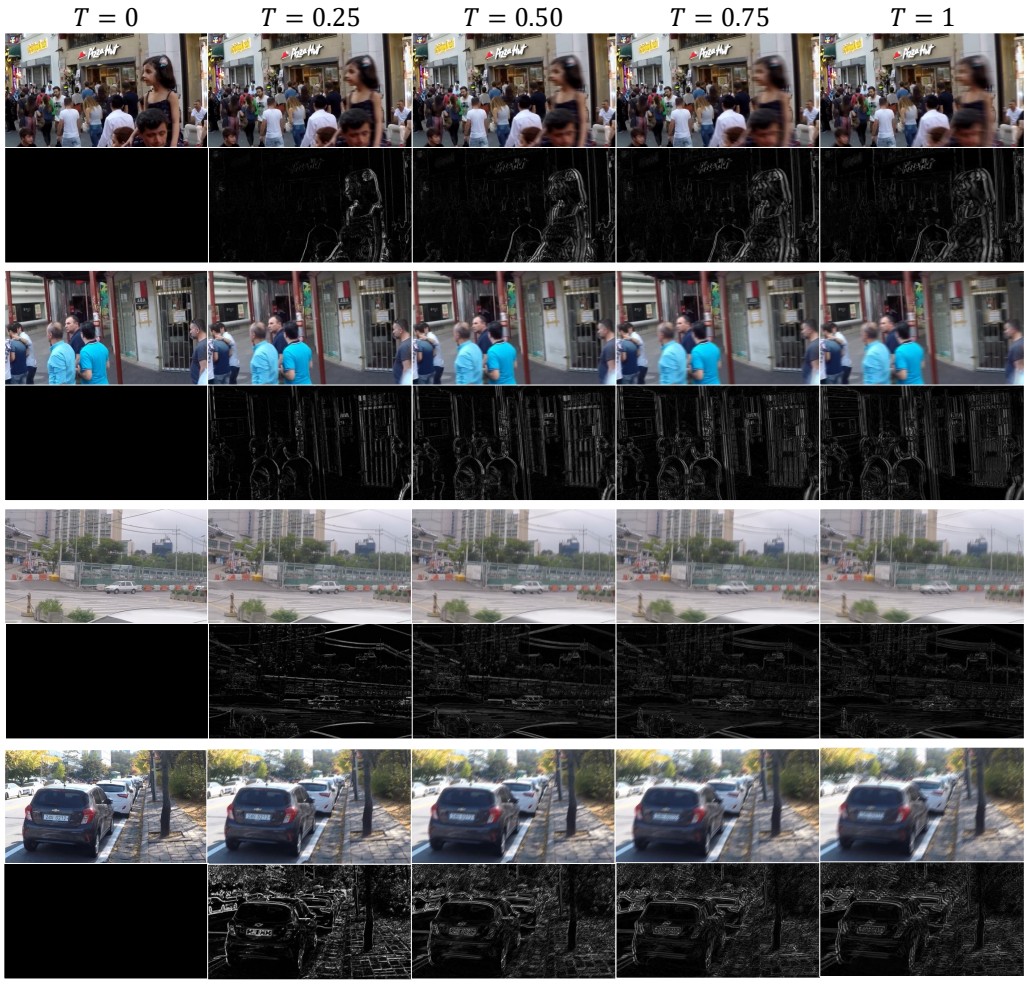

Figure 9: Visualizations of blur residual from GoPro [24] dataset

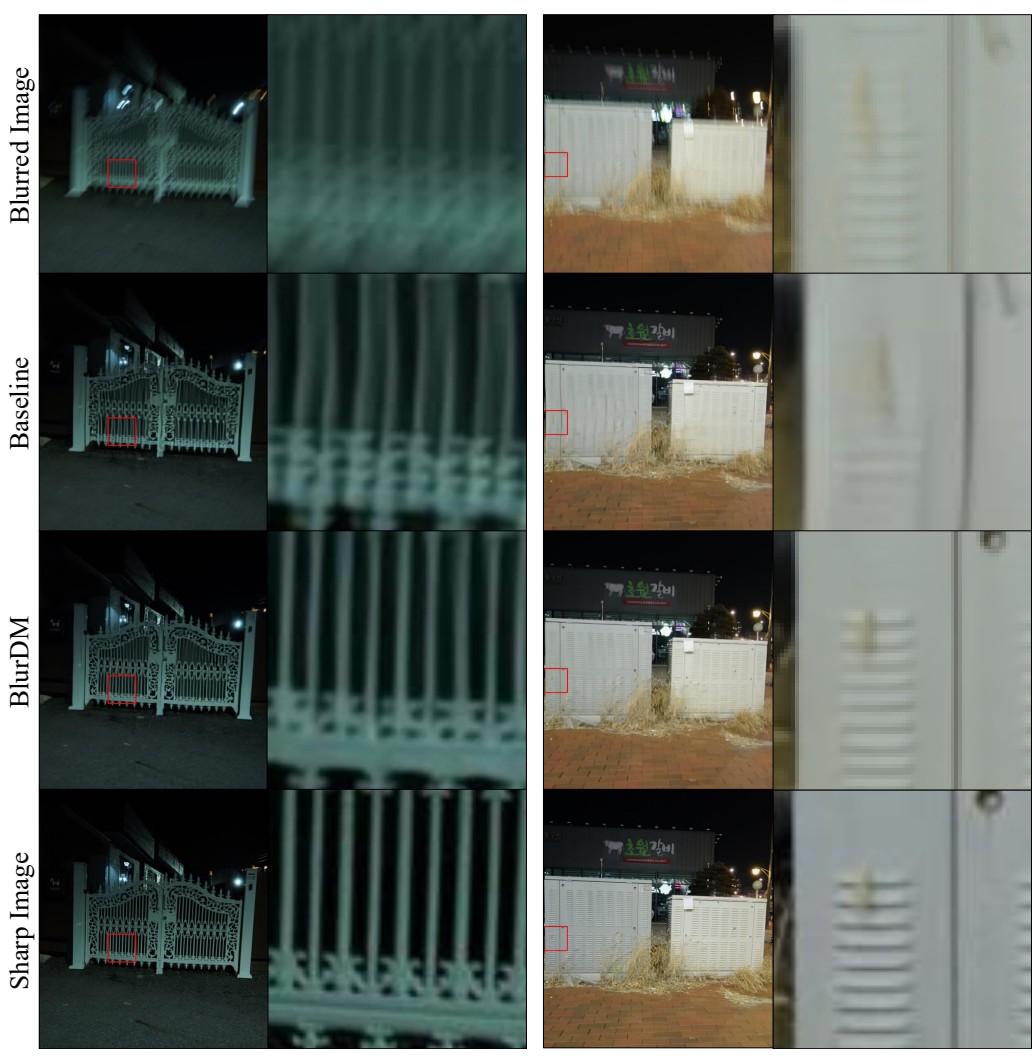

Figure 10: Qualitative results of MIMO-UNet [5] on the RealBlur-J [31] dataset.

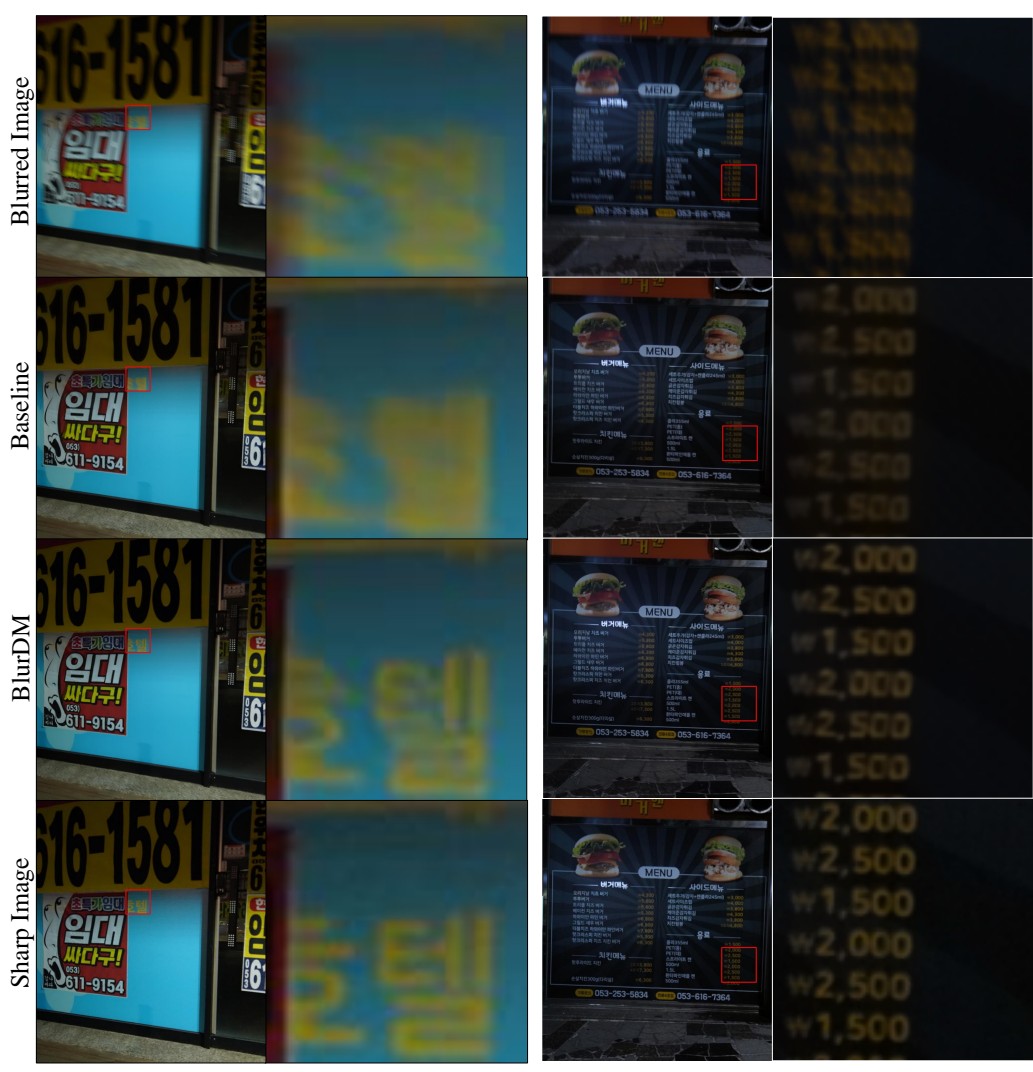

Figure 11: Qualitative results of Stripformer [40] on the RealBlur-J [31] dataset.

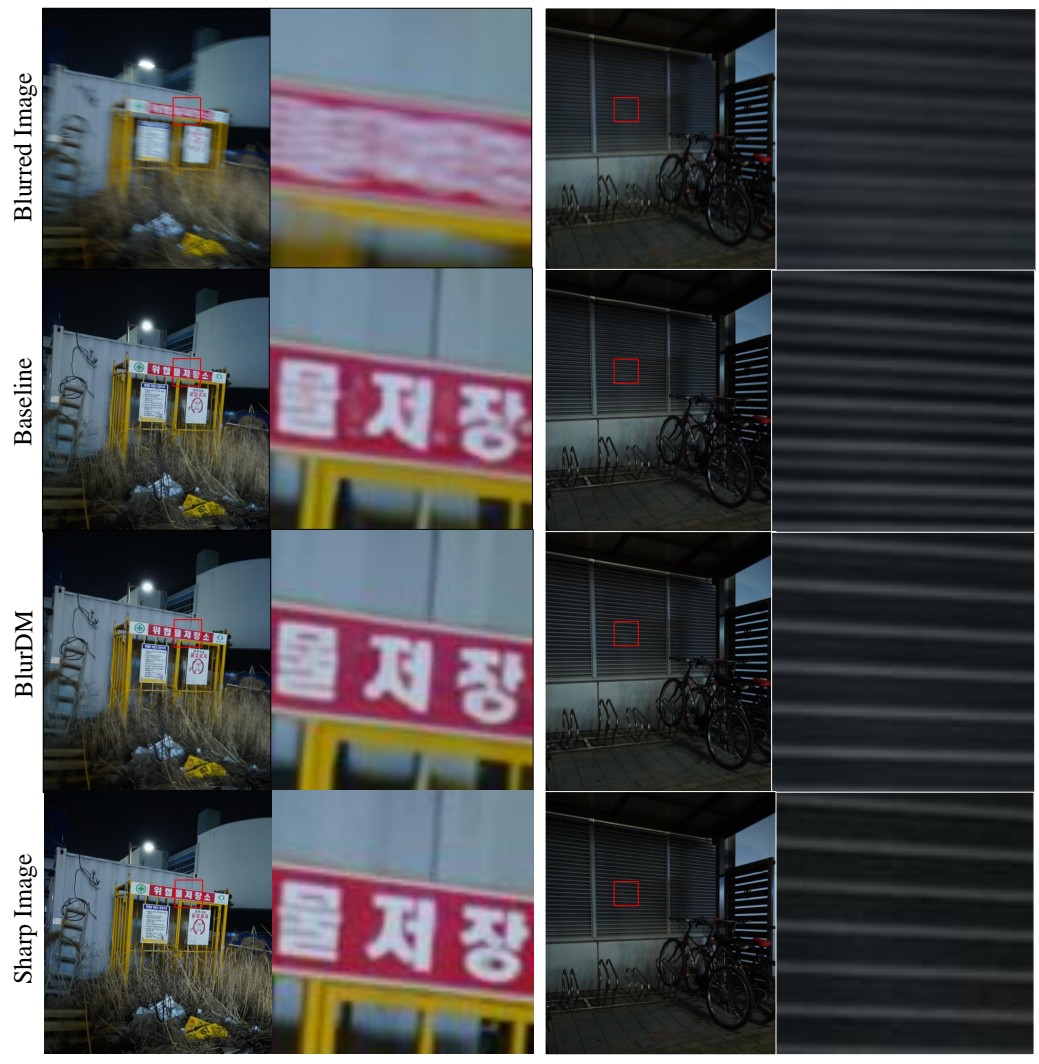

Figure 12: Qualitative results of FFTformer [13] on the RealBlur-J [31] dataset.

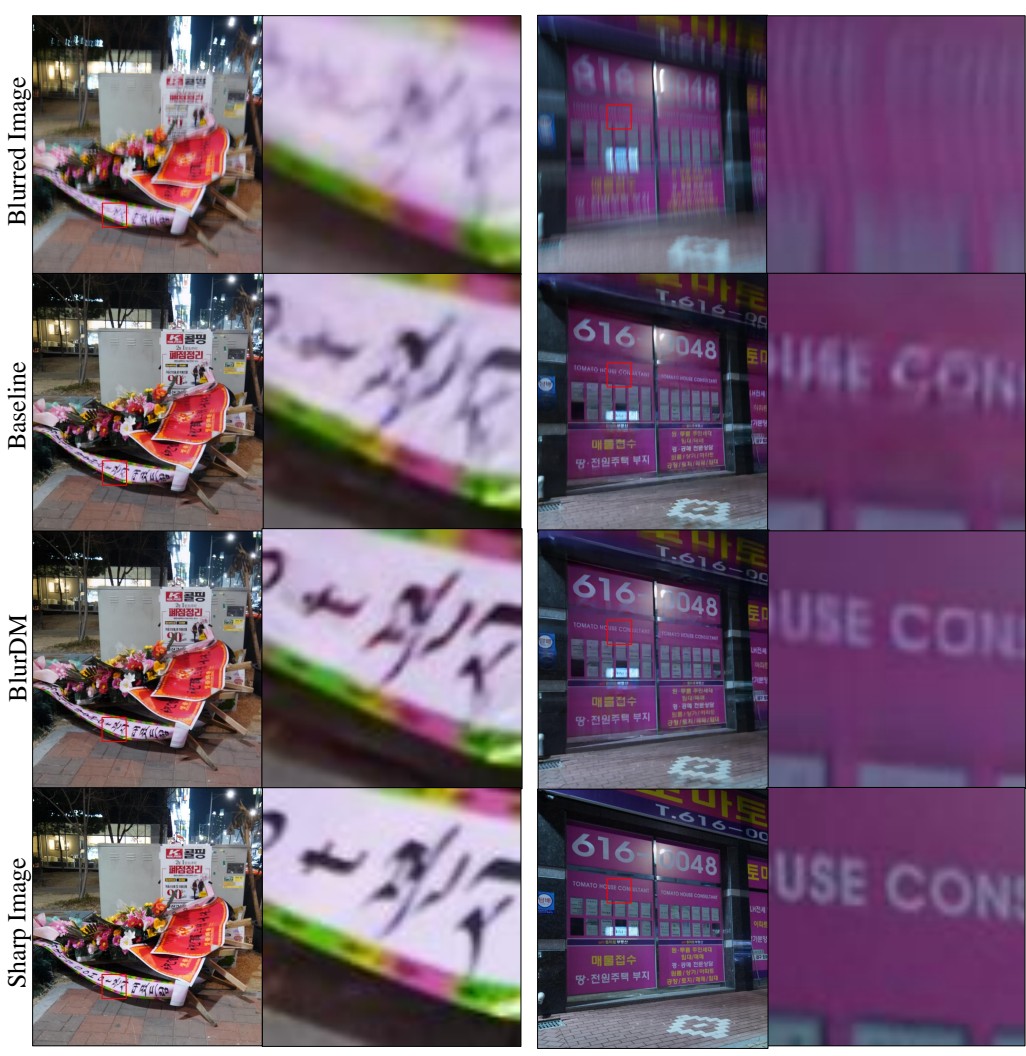

Figure 13: Qualitative results of LoFormer [21] on the RealBlur-J [31] dataset.

