# OpenReview forum: "BlurDM: A Blur Diffusion Model for Image Deblurring"
_NeurIPS.cc/2025/Conference — NeurIPS 2025 poster_

### Official Review · Reviewer_BwW8 · 2025-06-29

**Clarity:** 3
**Significance:** 3
**Originality:** 3
**Rating:** 4
**Confidence:** 3

**Summary:**

This work integrates the blur formation process into the diffusion framework by introducing a dual-diffusion scheme with forward and reverse stages. In the forward pass, both noise and blur are applied to a sharp image, while the reverse pass simultaneously performs denoising and deblurring to recover the original image. The proposed method, BlurDM, operates in the latent space to provide a flexible prior for image deblurring. Experimental results show that BlurDM consistently improves the performance of existing deblurring methods.

**Questions:**

Please address comments under weaknesses, and also the below pointers

1. While the introduction and formulation sections suggest that the method aims to incorporate natural blur formation into the diffusion process, the role of noise in naturally blurred images is neither discussed nor modeled. The presented dual noise and blur insertion is contradictory to real-world blur formation, where increased blur typically leads to reduced noise. It would strengthen the paper to acknowledge this distinction and clarify how diffusion-based modeling aligns with or diverges from natural blur formation processes.

2. In Table 1, the combination of MiMo-UNet and BlurDM achieves a PSNR of 32.93 on the GoPro dataset. However, in Tables 2 and 3, the same MiMo-UNet baseline with BlurDM reports a PSNR of only 32.28. The source of this discrepancy is not explained and warrants clarification.

**Ethical Concerns:**

["NO or VERY MINOR ethics concerns only"]

**Final Justification:**

Rebuttal has addressed all of my concerns, hence I am would like to retain my original rating.

**Limitations:**

As I mentioned under the questions, it'd good to clarify how diffusion-based modeling aligns with or diverges from natural blur formation processes

**Quality:**

3

**Strengths And Weaknesses:**

Strengths

1. Proposed method is able to significantly enhance the performance in terms of PSNR and SSIM compared to four different baseline architecture.
2. Proposed method presents the novel idea of inserting the blur formation into diffusion and showcase the advantages of doing so via experimental results and extensive evaluations.

Weaknesses

1. Although the method is built upon four existing architectures, its standing among prior works remains unclear. For instance, [17] reports that Uformer [42] outperforms their method, yet Uformer is neither directly compared nor included in enhancement or ablation studies. This omission is particularly notable given that [17] achieves performance levels close to the proposed method, as shown in Table 3. It would be informative to see how [42] compares in the same context.

2. Referring to Table 9 in [17], it is evident that PSNR tends to decrease as the number of diffusion steps increases. Therefore, for a fair PSNR-based comparison, consistent or reduced step counts should be maintained. Unfortunately, the number of steps used in the presented comparisons is not specified, making the reported improvements harder to interpret. Additionally, including LPIPS comparisons would help assess perceptual quality and reveal any trade-offs between perceptual fidelity and PSNR gains. Without this, the risk of over-optimizing solely for PSNR cannot be ruled out.

---

> ### Author Rebuttal · Authors · 2025-07-31
>
> We thank the reviewer for recognizing the novelty of our approach and its significant performance gains shown in experiments. We greatly appreciate these encouraging comments. We address each concern in detail below and will revise the paper accordingly.
>
> **Weakness 1: About the standing among prior works and the performance of BlurDM to Uformer**
>
> Regarding the concern about the standing among prior works, we would like to clarify that our proposed method, BlurDM, is a plug-and-play framework with strong generalization ability, designed to integrate into arbitrary image deblurring backbones. To demonstrate this, we conducted comparisons in Table 1 against four widely recognized deblurring methods, including FFTformer, which was the state-of-the-art at the time of paper submission. These comparisons clearly validate that our framework can effectively enhance strong baseline architectures.
>
> Additionally, as you suggested, we conduct experiments using Uformer as the deblurring backbone. As shown in Table G, our method consistently improves Uformer’s performance on the GoPro and HIDE datasets, verifying the effectiveness and robustness of our proposed approach across different architectures.
>
> | **Backbone**         | **GoPro** | **HIDE** |
> |----------------------|-----------|----------|
> | Uformer              | 32.97     | 30.83    |
> | Uformer + BlurDM     | 33.17     | 30.97    |
>
> **Table G**: Performance comparison between Uformer and Uformer + BlurDM on GoPro and HIDE datasets.
>
> **Weakness 2: Performance comparison of the number of steps used in RDDM and our method**
>
> To ensure a fair evaluation in Table 3, we set the diffusion steps to five for all models. For additional insights, Table H presents a comparison of RDDM and BlurDM under varying numbers of diffusion steps, where PSNR and LPIPS are used to assess performance. RDDM achieves its peak PSNR of 32.08 dB at step 4 and the best LPIPS score of 0.0121 at step 10. In comparison, BlurDM achieves a higher peak PSNR of 32.28 dB at step 5 and a lower LPIPS score of 0.0113 at step 10. These results demonstrate that BlurDM consistently outperforms RDDM in terms of both distortion-based (PSNR) and perceptual (LPIPS) quality metrics.
>
> | **Method** | **2**       | **4**       | **5**       | **6**       | **8**       | **10**      |
> |------------|-------------|-------------|-------------|-------------|-------------|-------------|
> |            | PSNR↑ / LPIPS↓ | PSNR↑ / LPIPS↓ | PSNR↑ / LPIPS↓ | PSNR↑ / LPIPS↓ | PSNR↑ / LPIPS↓ | PSNR↑ / LPIPS↓ |
> | RDDM       | 32.07 / 0.0130 | 32.08 / 0.0131 | 32.03 / 0.0125 | 32.01 / 0.0122 | 32.02 / 0.0122 | 32.02 / 0.0121 |
> | BlurDM     | 32.21 / 0.0116 | 32.27 / 0.0114 | **32.28** / 0.0114 | 32.22 / 0.0115 | 32.24 / 0.0113 | 32.24 / **0.0113** |
>
> **Table H**: PSNR↑ / LPIPS↓ under different step counts on the GoPro dataset. BlurDM shows superior performance in both fidelity and perceptual quality.
>
>
> **Question 1: The role of noise in blurred images**
>
> We would like to clarify that the “noise” in our dual diffusion framework does not refer to sensor noise in captured images. Instead, motivated by recent studies (Lines 108–123) highlighting the strong generative capabilities of diffusion models in reconstructing missing high-frequency details, we leverage the noise diffusion process to stochastically synthesize these lost high-frequency details in heavily blurred regions. This is conceptually different from modeling physical sensor noise. The discussion above will be incorporated into the paper.
>
> **Question 2: The performance of MiMo-UNet baseline reported in Tables 2 and 3**
>
> To ensure a fair comparison, we adopt different training schedules for different experimental purposes. Specifically, in Table 1, MIMO-UNet + BlurDM is trained with 3000 epochs for Stage 1, 500 for Stage 2, and 3000 for Stage 3. In contrast, Tables 2 and 3 use a standardized schedule of 1000/500/1000 epochs for the three stages to ensure consistency across methods in the ablation and component analysis. We will clarify it in the paper.

---

> > ### Comment · Reviewer_BwW8 · 2025-08-08
> > **Response to the rebuttal**
> >
> > Sorry for the late response. I would like to thank authors for the clarifications provided in the rebuttal which have satisfactorily addressed my concerns.

---

> > > ### Author Response · Authors · 2025-08-09
> > >
> > > Thank you for your comment, positive feedback, and support!

---

> ### Author Response · Authors · 2025-08-05
>
> Dear reviewer,
>
> Thank you for the comments on our paper.
>
> We have submitted the responses to your comments. Please let us know if you have additional questions so that we can address them during the discussion period.
>
> Thank you!
>
> Best,
>
> Authors

---

> ### Author Response · Authors · 2025-08-06
>
> We appreciate your review. We hope our responses have addressed your concerns. As the discussion deadline is approaching, please let us know if you have further questions after reading our rebuttal. We aim to address all potential issues during the discussion period and hope that you will consider raising the score.
>
> Thank you!
>
>
> Best,
>
> Authors

---

> ### Author Response · Authors · 2025-08-08
>
> Dear Reviewer,
>
> With only one day remaining in the discussion phase, we welcome any additional feedback you may have so that we can address all remaining concerns.
>
> Authors

---

> ### Comment · Area_Chair_FBDe · 2025-08-08
>
> Hi Reviewer BwW8,
>
> This is a gentle reminder to participate in the discussion with the authors regarding their rebuttal. Your input at this stage is important and appreciated.
>
> Best, AC

---

### Official Review · Reviewer_AFXA · 2025-06-30

**Clarity:** 2
**Significance:** 3
**Originality:** 3
**Rating:** 4
**Confidence:** 3

**Summary:**

This work offers a unique image deblurring approach named the Blur Diffusion Model (BlurDM), which incorporates blur process into the diffusion model. Its primary advances are: 1. a dual-diffusion forward scheme; 2. a dual denoising and deblurring formulation; and 3. the incorporation of BlurDM into the latent space of current deblurring architectures. Experiments demonstrate that BlurDM can increase the performance of deblurring algorithms on a variety of benchmark datasets.

**Questions:**

1. Why aren't there direct comparisons to existing diffusion-based deblurring methods?  Such comparisons would assist to isolate the benefits of the proposed blur-aware diffusion formulation.

2. Can the authors explain the increased complexity brought on by BlurDM in terms of training time and convergence behavior?  The three-stage pipeline is successful but may be difficult to replicate; extra training information would be beneficial.

**Ethical Concerns:**

["NO or VERY MINOR ethics concerns only"]

**Final Justification:**

The authors have provided detailed responses that address my concerns. I am satisfied with their clarifications and will maintain my original score.

**Limitations:**

Yes

**Paper Formatting Concerns:**

No major formatting issues were found.

**Quality:**

3

**Strengths And Weaknesses:**

Strengths:

1. Theoretical derivations are robust, with a thorough definition of forward and reverse processes, and deterministic DDIM-style sampling assures efficiency and efficacy.
2. BlurDM is modular and flexible, acting as a prior module that can be integrated into existing deblurring networks.
3. Extensive testing with four datasets and four approaches show consistent performance benefits at low computational expense.

Weaknesses:

1. The technique's application in more complicated real-world circumstances is limited because it is mainly designed for motion blur and cannot handle defocus blur.
2. Since there is no direct comparison with current diffusion-based deblurring models, it is challenging to determine the actual value of the suggested blur-aware formulation.
3. The entire framework's complexity—which includes many modules and multi-stage training—increases implementation costs and diminishes clarity.

---

> ### Author Rebuttal · Authors · 2025-07-31
>
> We thank the reviewer for recognizing the robustness of our theoretical derivations, the modularity and flexibility of BlurDM, and the consistent performance gains demonstrated across extensive experiments. We greatly appreciate these encouraging comments. We address each concern in detail below and will revise the paper accordingly
>
> **Weakness 1: Generalizability of the BlurDM for defocus blur**
>
> We appreciate the comment regarding the generalizability. While we agree that this paper primarily focuses on motion deblurring, we wish to emphasize that motion blur stands as a persistently challenging and unresolved research topic in the literature of image restoration. It is evidenced that recent impactful works such as MIMO-UNet, Stripformer, FFTformer, and LoFormer focus exclusively on motion deblurring, highlighting the importance of addressing motion blur in this field. Additionally, although BlurDM is specifically designed to address motion blur, it can be flexibly integrated into general-purpose deblurring backbones capable of handling both motion and defocus blur. This allows BlurDM to provide motion-aware guidance that effectively enhances motion deblurring performance.
>
> As a future direction, we plan to extend BlurDM by incorporating the physical characteristics of defocus blur into the diffusion process, similar to what we did in Equation (1) for motion blur. It enables a unified prior that addresses more complex and mixed blur types.
>
> **Weakness 2 and Question 1: Direct comparison with current diffusion-based deblurring models**
>
> As mentioned in Lines 62–64, prior studies have shown that applying diffusion models directly in the image space for deblurring often struggles to recover fine details due to the inherent stochasticity of the generative process. Moreover, image-space diffusion is computationally expensive and memory-intensive, particularly for high-resolution datasets like GoPro and RealBlur. Therefore, we focus on latent-space diffusion and conduct a fair comparison by evaluating different diffusion algorithms, including DDPM and RDDM, within the latent space using the same deblurring backbone, as shown in Table 3.
>
> Additionally, we directly compare BlurDM with two recent diffusion-based deblurring methods, including HI-Diff [4] and RDDM [17], on the GoPro dataset, as shown in Table E. HI-Diff applies DDPM to the deblurring network in the latent space, whereas RDDM performs the residual diffusion process in the image space. BlurDM consistently achieves better performances in PSNR and SSIM under comparable parameter counts and FLOPs. Moreover, as a plug-and-play module, BlurDM can be flexibly integrated into various backbone architectures. This demonstrates not only its superior performance but also its strong generalizability across different network designs.
>
> | **Method**              | **Venue**     | **PSNR** | **SSIM** | **Params (M)** | **FLOPs (G)** |
> |-------------------------|---------------|----------|----------|----------------|----------------|
> | HI-Diff                 | NeurIPS’ 23    | 33.33    | 0.955    | 23.99          | 125.47         |
> | RDDM                    | CVPR’ 24       | 32.40    | 0.963    | 15.49          | 134.20         |
> | BlurDM (Stripformer)    | -             | 33.53    | 0.966    | 24.33          | 174.18         |
> | BlurDM (FFTformer)      | -             | 34.34    | 0.970    | 18.66          | 135.69         |
> | BlurDM (LoFormer)       | -             | 33.70    | 0.967    | 19.08          | 56.35          |
>
> **Table E**: Comparison of different deblurring methods in terms of PSNR, SSIM, number of parameters, and FLOPs.
>
>
> **Weakness 3 and Question 2: Training cost of the three-stage training strategy**
>
> In Table F, we report the training complexity introduced by BlurDM in terms of training time and convergence behavior, using MIMO-UNet as the backbone. The default number of training epochs for MIMO-UNet is 3, 000, denoted as “Baseline 1”. Increasing the training epochs to 6, 000, denoted as “Baseline 2”, results in only a marginal improvement of 0.07 dB in PSNR. For BlurDM, we experiment with different training configurations, including using 1, 500 or 3, 000 epochs for Stage 1 and Stage 3, with 500 epochs for Stage 2. Both “BlurDM 1” and “BlurDM 2” outperform Baseline 2 while requiring less training time, demonstrating the effectiveness of the proposed three-stage training strategy. Finally, we adopt “BlurDM 3” as our final model, which increases training time by only 8% while achieving a 0.42 dB improvement in PSNR compared to “Baseline 2”.
>
> As for convergence behavior, according to our observation of the learning curves, the optimization process does not suffer from any convergence issue. Furthermore, we will release all our code upon the paper’s acceptance to ensure reproducibility.
>
> | **Method**   | **Training epoch**                                              | **Training time** | **PSNR** |
> |--------------|------------------------------------------------------------------|-------------------|----------|
> | Baseline 1   | 3000                                                             | 66.7 h            | 32.44    |
> | Baseline 2   | 6000                                                             | 133.4 h           | 32.51    |
> | BlurDM 1     | 1500 (Stage 1) + 500 (Stage 2) + 1500 (Stage 3)                  | 70.7 h            | 32.62    |
> | BlurDM 2     | 3000 (Stage 1) + 500 (Stage 2) + 1500 (Stage 3)                  | 104.1 h           | 32.71    |
> | BlurDM 3     | 3000 (Stage 1) + 500 (Stage 2) + 3000 (Stage 3)                  | 141.4 h           | 32.93    |
>
> **Table F**: Comparison of training cost and performance between baseline and BlurDM.

---

> ### Author Response · Authors · 2025-08-05
>
> Dear reviewer,
>
> Thank you for the comments on our paper.
>
> We have submitted the responses to your comments. Please let us know if you have additional questions so that we can address them during the discussion period.
>
> Thank you!
>
> Best,
>
> Authors

---

> > ### Comment · Reviewer_AFXA · 2025-08-05
> >
> > Thank you for the clarification provided in the rebuttal. I appreciate the authors' detailed responses, which have satisfactorily addressed my main concerns. I am maintaining my original score.

---

> > > ### Author Response · Authors · 2025-08-05
> > >
> > > Thank you for your positive feedback and support!

---

### Official Review · Reviewer_EKwB · 2025-07-01

**Clarity:** 3
**Significance:** 3
**Originality:** 3
**Rating:** 5
**Confidence:** 4

**Summary:**

This paper proposes BlurDM, a diffusion model for dynamic scene deblurring that integrates the physical process of motion blur formation (specifically, its temporal integration over exposure time) directly into the diffusion steps. It derives a dual denoising and deblurring formulation, implicitly estimating the blur residual while progressively restoring a clear image. A latent variant, Latent BlurDM, is also developed for efficient integration. Experiments across multiple datasets demonstrate competitive deblurring performance, particularly in recovering high-frequency details.

**Questions:**

My major confusion is about the training of the blur residual estimator. Specifically, as can be seen in Eq. (34), the blur residual estimator is supervised by $e_t$, which is theoretically the blur residual from $\alpha_{t-1}$ to $\alpha_t$. However, it is not clear how to obtain it in practice, at least to me. Please make it more clearly clarified, if I miss something.

**Ethical Concerns:**

["NO or VERY MINOR ethics concerns only"]

**Final Justification:**

The authors have addressed my concern. So I keep my initial rating.

**Limitations:**

The authors have discussed the limitations of their method.

**Paper Formatting Concerns:**

There is no major formatting issue.

**Quality:**

3

**Strengths And Weaknesses:**

### Strengths

1. The core idea of embedding continuous blur dynamics into the diffusion process is theoretically well-motivated and novel. It captures the essence of deblurring by explicitly linking blur formation and diffusion, offering a more principled approach than directly applying existing diffusion models.

2. Experimental results are promising, especially in restoring high-frequency details.

### Weaknesses

There is no major issue for me.

---

> ### Author Rebuttal · Authors · 2025-07-31
>
> We thank the reviewer for recognizing our idea as theoretically well-motivated and novel, and for acknowledging the promising experimental results. We greatly appreciate these encouraging comments and address the concern in detail below.
>
> **Questions 1: Supervision of blur residual $e_t$ during training**
>
> As detailed in Lines 512–525, we do not explicitly supervise the individual blur residuals $e_t$ during training. Instead, BlurDM is optimized using gradients derived from the reconstruction loss between the deblurred result $I^{\theta}_0$ and the ground-truth sharp image $I^0$ through Equation (36).  Since BlurDM is a physics-guided module that estimates blur residuals based on the principles of blur formation, as formulated in Equation (12), it can effectively estimate these residuals without requiring ground-truth supervision for intermediate blur residuals. Despite the absence of intermediate ground-truth supervision, the deblurred results at different time steps shown in Figure 5 demonstrate that BlurDM effectively captures blur residuals throughout the reverse process.

---

> ### Author Response · Authors · 2025-08-05
>
> Dear reviewer,
>
> Thank you for the comments on our paper.
>
> We have submitted the responses to your comments. Please let us know if you have additional questions so that we can address them during the discussion period.
>
> Thank you!
>
> Best,
>
> Authors

---

> > ### Comment · Reviewer_EKwB · 2025-08-05
> >
> > Thanks for your response, which has addressed my concern.

---

> > > ### Author Response · Authors · 2025-08-05
> > >
> > > Thank you for your comments and support!

---

### Official Review · Reviewer_a5f6 · 2025-07-01

**Clarity:** 2
**Significance:** 3
**Originality:** 3
**Rating:** 4
**Confidence:** 3

**Summary:**

This paper presents BlurDM, diffusion-based framework for image deblurring that models the forward degradation as a combination of noise and motion blur accumulation, reflecting the `physical' process during exposure. BlurDM employs a dual diffusion process to estimate a sharp prior in the latent space, which then guides a separate restoration network for final deblurring. The framework is trained in three stages: 1) pretraining a Sharp Encoder and Prior Fusion Module, 2) training BlurDM to generate sharp priors from blurred inputs, and 3) jointly optimizing all components. BlurDM is compatible with existing deblurring architectures, and consistently improves performance across various benchmarks with minimal computational overhead.

**Questions:**

1. It would be valuable for the authors to conduct ablation studies by omitting individual stages to quantify each stage’s contribution and evaluate the potential risk of error propagation throughout the pipeline.

2. Real-world scenarios often involve diverse and complex blur patterns, rather than averaging interpolated blurs. It would strengthen the paper to evaluate BlurDM under varying conditions—such as different blur intensities and in the presence of noise or JPEG compression—to better demonstrate its practical robustness.

3. Regarding the evaluation using Z^s: Since Z^s can be an upper bound on achievable deblurring performance, it would be insightful to provide further analysis or comparison to clarify its role and justify its use within the framework.

**Ethical Concerns:**

["NO or VERY MINOR ethics concerns only"]

**Final Justification:**

The rebuttal has addressed many of my concerns; however, the reviewer still finds the main validation somewhat uncertain.
Nonetheless, the additional ablation studies have effectively resolved the primary concern. the reviewer will therefore raise the score.
Please include all ablation studies and the related discussion in the final version.

**Limitations:**

No.

**Quality:**

3

**Strengths And Weaknesses:**

Strengths:

1) BlurDM is grounded in the physical modeling of motion blur, offering intuitive insights into the deblurring process.

2) Experimental results clearly demonstrate the effectiveness of the proposed dual diffusion strategy.

Weaknesses:

1) The validity of the Equations (1) to (3) within the latent space is questionable. While noise is explicitly added, the treatment of blur as simply added lacks theoretical justification or explanation.

2) Further evidence or analysis is needed to confirm that motion blur in the latent space indeed converges to a Gaussian distribution, as assumed in the proposed model.

---

> ### Author Rebuttal · Authors · 2025-07-31
>
> We thank the reviewer for recognizing the strength of our core idea and the clear effectiveness of the proposed dual diffusion strategy in experiments. We greatly appreciate these encouraging comments. We address each concern in detail below and will revise the paper accordingly.
>
> **Weaknesses 1: Theoretical justification of latent-space BlurDM**
>
> Previous works, such as Stable Diffusion [a], HI-Diff [15], and Swin-diff [16], have demonstrated the effectiveness and efficiency of shifting the diffusion process from the image space to the latent space by leveraging latent space encoders such as VAEs and CNNs. Inspired by these methods, we employ sharp and blur encoders to learn latent representations for performing the diffusion process in the latent space through a three-stage training strategy. Specifically, in the forward diffusion process, instead of adding blur step by step to images, we derive a one-step blur addition process, as described in Appendix A.1. This process demonstrates that the final blurred and noisy image can be obtained by directly adding noise to the blurred image provided in the original dataset. In the reverse generation process, we utilize Equation (12) to iteratively remove blur features, following the principles of the blur formation process in the forward diffusion stage. Moreover, to demonstrate that our latent-space diffusion process performs deblurring in an iterative manner, we present deblurred results at different time steps in Figure 5, confirming that blur residuals are effectively addressed in the latent domain.
>
> Reference:
>
> [a] Robin Rombach, Andreas Blattmann, Dominik Lorenz, Patrick Esser, Björn Ommer. High-Resolution
> Image Synthesis with Latent Diffusion Models. In CVPR, 2022.
>
> **Weaknesses 2: Question about the convergency of motion blur to the Gaussian distribution**
>
> As described in Equation (6), we model motion blur as a mean shift of the standard Gaussian diffusion process, where the mean is shifted by $\frac{1}{\alpha_T} \sum_{t=1}^{T}e_t$ that captures the degradation characteristics introduced by motion blur. This mean-shift interpretation is conceptually aligned with the idea presented in Figure 1 of the RDDM paper [17].  However, as mentioned in Lines 118–123, RDDM models the blur residual by "linearly'' computing the difference between the blurred and sharp image via subtraction.  In contrast, we drive a dual denoising and deblurring formulation that follows the principles of the blur formation process to align with the natural characteristics of blur, typically caused by a "convolutional'' process.
>
> **Question 1 and Question 3: Effectiveness of the three-stage training strategy**
>
> In Table D, we present ablation studies to evaluate the effectiveness of the three-stage training strategy. We use MIMO-UNet as the backbone, and all models are trained for 1000 epochs to ensure a fair comparison. “Net1” denotes the baseline deblurring performance without incorporating BlurDM. In “Net2”, we directly use the sharp prior $Z^S$ in BlurDM, serving as an upper bound for the achievable deblurring performance. In “Net3”, we jointly optimize BlurDM and the deblurring model without pre-training through Stage 1 and Stage 2, serving as a baseline for a purely data-driven approach. In “Net4” and “Net5”, after completing pre-training in Stage 1, we apply either Stage 2 or Stage 3 alone to optimize BlurDM and the deblurring model. In “Net6”, we adopt the complete three-stage training strategy, which achieves the best performance compared to the alternative combinations used in “Net3” to “Net5”.
>
> | **Model** | **Stage 1** | **Stage 2** | **Stage 3** | **PSNR**             |
> |-----------|-------------|-------------|-------------|----------------------|
> | Net1      |             |             |             | 31.78 (Baseline)     |
> | Net2      | ✓           |             |             | 32.69 (Upper bound)  |
> | Net3      |             |             | ✓           | 31.80                |
> | Net4      | ✓           |             | ✓           | 32.01                |
> | Net5      | ✓           | ✓           |             | 31.95                |
> | Net6      | ✓           | ✓           | ✓           | **32.28**            |
>
> **Table D**: Ablation study of each training stage in BlurDM.
>
> **Question 2: Evaluation of BlurDM on real-world blur scenarios under varying conditions**
>
> In Table 1 of our main paper, we evaluate BlurDM on the challenging real-world dataset RealBlur, which contains naturally captured blur under low-light conditions, often accompanied by severe blur intensity and sensor noise. BlurDM consistently improves the performance of existing methods on the RealBlur dataset, achieving gains of up to 1.24 dB on RealBlur-J and 1.16 dB on RealBlur-R in PSNR, thereby verifying the robustness of our approach under diverse degradation conditions.

---

> ### Author Response · Authors · 2025-08-05
>
> Dear reviewer,
>
> Thank you for the comments on our paper.
>
> We have submitted the responses to your comments. Please let us know if you have additional questions so that we can address them during the discussion period.
>
> Thank you!
>
> Best,
>
> Authors

---

> > ### Comment · Reviewer_a5f6 · 2025-08-05
> >
> > The reviewer thanks the authors for their rebuttal and, in particular, appreciates the additional experiments demonstrating the effectiveness of the three-stage training strategy.

---

> > > ### Author Response · Authors · 2025-08-06
> > >
> > > We appreciate your review and feedback. In addition to the new experiments on the three-stage training strategy, we hope our responses address all of your concerns in the review. Please let us know if you have any further questions after reading our rebuttal and the derivation to validate latent BlurDM. We aim to address all potential issues during this discussion period and hope you will consider raising the score.
> > >
> > > Thank you!
> > >
> > > Best,
> > >
> > > Authors

---

> ### Author Response · Authors · 2025-08-05
>
> Thank you for the insightful review and positive feedback.
>
> To further address your concern about weakness 1, we provide the theoretical derivation of latent BlurDM.
> Let  $z_t = E(I_t)$, where $I_t$ is the blurred image at time step  $t$, $E(\cdot) $ is the encoder, and $z_t$ is the corresponding latent feature.
> Using the first-order Taylor expansion of the encoder around $I_{t-1}$, we have:
>
> $
> z_t \approx z_{t-1} + J_E(I_{t-1})(I_t - I_{t-1}),
> $
>
> where \$J_E(I_{t-1})$ is the Jacobian of the encoder evaluated at $I_{t-1}$.
> From Equation (3) in the paper, it can be derived
>
>
> $I_t - I_{t-1} = \left( \frac{\alpha_{t-1}}{\alpha_t} - 1 \right) I_{t-1} + \frac{1}{\alpha_t} e_t + \beta_t \epsilon_t.$
>
> Plugging the above equation into the Taylor expansion yields:
>
> $
> z_t \approx\  z_{t-1}+ J_E(I_{t-1}) \left( \frac{\alpha_{t-1}}{\alpha_t} - 1 \right) I_{t-1} + \frac{1}{\alpha_t} J_E(I_{t-1}) e_t + \beta_t J_E(I_{t-1}) \epsilon_t.
> $
>
> Since $\frac{\alpha_{t-1}}{\alpha_t} \approx 1$ when the exposure intervals are sufficiently small, the expression simplifies to:
>
> $z_t \approx \left( \frac{\alpha_{t-1}}{\alpha_t} \right) z_{t-1} + \frac{1}{\alpha_t} e_t^\theta + \beta_t \epsilon_t^\theta,$
>
> where $\theta$ denotes the learnable parameters as shown in Equation (12). $e_t^\theta = J_E(I_{t-1}) e_t $ and $ \epsilon_t^\theta = J_E(I_{t-1}) \epsilon_t $ represent the blur residual term and the noise term projected into the latent space, respectively. This demonstrates that the noise and blur terms in the latent BlurDM can be characterized and learned in the same way as the two corresponding terms in Equation (3).

---

> > ### Comment · Reviewer_a5f6 · 2025-08-08
> > **Uncertainty of the equation**
> >
> > Thank you for the insightful comments. However, another concern arises. Since the encoder encoder (E) may not follow a linear relationship, the first-order Taylor expansion might not accurately approximate $ z_t \approx z_{t-1} + J_E(I_{t-1})(I_t - I_{t-1})$. Could the authors clarify the method used to obtain this approximation?

---

> > > ### Author Response · Authors · 2025-08-08
> > > **Justification of the First-Order Taylor Approximation in Our Setting**
> > >
> > > Dear Reviewer,
> > >
> > > We appreciate your further feedback. A Taylor expansion can be used to approximate any differentiable nonlinear function. When the difference $(I_t - I_{t-1})$ is sufficiently small---such as in short exposure intervals---the higher-order terms $(I_t - I_{t-1})^n$ for $n \geq 2$ become negligible. Consequently, the nonlinear encoder function $E(\cdot)$ can be effectively approximated using the first-order Taylor expansion as follows:
> > >
> > > $
> > > z_t \approx z_{t-1} + J_E(I_{t-1})(I_t - I_{t-1}),  \text{where} \quad z_t = E(I_t), \quad z_{t-1} = E(I_{t-1})
> > > $
> > >
> > > Please feel free to let us know if you have any further questions.
> > >
> > > Thank you.
> > >
> > >
> > > Best,
> > >
> > > Authors

---

> > > > ### Author Response · Authors · 2025-08-09
> > > >
> > > > Dear Reviewer,
> > > >
> > > > We trust we have addressed your concerns and remain available to respond to any further comments in the final hours of the discussion phase. If you are satisfied with our responses, we would appreciate your consideration in raising the score.
> > > >
> > > > Authors

---

> ### Author Response · Authors · 2025-08-08
>
> Dear Reviewer,
>
> With only one day remaining in the discussion phase, we welcome any additional feedback you may have so that we can address all remaining concerns.
>
> Authors

---

### Official Review · Reviewer_zo6Q · 2025-07-02

**Clarity:** 1
**Significance:** 1
**Originality:** 2
**Rating:** 2
**Confidence:** 3

**Summary:**

The paper proposes BlurDM, a diffusion-based model for image deblurring that integrates a dual noise and blur diffusion process to align with the physical principles of motion blur formation. The authors introduce a blur residual estimator and a noise estimator to reverse the degradation process, alongside a Prior Fusion Module (PFM) to enhance existing deblurring networks. Experiments on four datasets (GoPro, HIDE, RealBlur-J/R) demonstrate consistent PSNR/SSIM improvements over baseline methods.

**Questions:**

1. Why is BlurDM’s blur residual estimator superior to RDDM’s additive residuals? Provide theoretical or empirical comparisons.
2. How does BlurDM scale to ultra-high-resolution images? Report metrics (e.g., memory, speed) for 4K inputs.
3. Could the framework handle hybrid blur (e.g., motion + defocus)? Address limitations in Section 5 more concretely.
4. Why omit statistical significance tests? Even simple bootstrapped confidence intervals would strengthen claims.

**Ethical Concerns:**

["NO or VERY MINOR ethics concerns only"]

**Limitations:**

The authors acknowledge BlurDM’s limitation to motion blur but overlook broader issues:
1. Ethical risks: Potential misuse for deblurring sensitive/private content (e.g., surveillance footage).
2. Computational cost: No comparison to lightweight deblurring methods (e.g., CNNs) for edge devices.

**Paper Formatting Concerns:**

None. The paper adheres to NeurIPS formatting guidelines.

**Quality:**

2

**Strengths And Weaknesses:**

Strengths:
1. Novelty:The dual diffusion process (noise + blur) is a creative adaptation of diffusion models for deblurring, addressing the structured nature of motion blur.
2. Technical Rigor:Detailed derivations (Section 3, Appendix A) justify the dual diffusion framework and reverse process.
3. Practical Integration:BlurDM’s latent-space implementation minimizes computational overhead (Table 4), making it feasible for deployment.

Weaknesses:
1. Incremental Contribution:The core idea (diffusion for deblurring) builds heavily on prior work (e.g., DDPM, RDDM). The dual diffusion process, while novel, offers modest gains (~0.3–0.8 dB PSNR in Table 1).
2. Limited Generalizability:Focused solely on motion blur; performance on defocus blur (Section 5) or other degradation types is unexplored.
3.No analysis of real-world edge deployment (latency, power consumption).

---

> ### Author Rebuttal · Authors · 2025-07-31
>
> We sincerely thank the reviewer for recognizing BlurDM’s novelty, technical rigor, and practical integration. We greatly appreciate these encouraging comments. We address each concern in detail below and will revise the paper accordingly.
>
> **Weaknesses 1 and Question 1: Fundamental Differences between DDPM, RDDM, and our BlurDM**
>
> As explained in Lines 35–45 and 108–117 of the paper, motion blur results from a continuous exposure process during image capture, where blur intensity accumulates progressively along a motion trajectory. This leads to highly structured and directionally consistent patterns, which differ significantly from the random noise perturbations modeled by conventional diffusion-based frameworks like DDPM.
>
> Previous works [3, 4, 15, 30] that directly apply DDPM to image deblurring neglect this critical physical aspect of blur formation, thereby limiting their capacity to model the real-world blur formation process and ultimately resulting in suboptimal deblurring performance.
>
> As mentioned in Lines 118–123, RDDM [17] models the blur residual by "linearly'' computing the difference between the blurred and sharp image via subtraction, which is inconsistent with real-world blur formation, typically caused by a "convolutional'' process.
>
> In contrast, we drive a dual denoising and deblurring formulation in Equation (4) that adheres to the principles of the blur formation process, aligning with the natural characteristics of blur. In Table 3, BlurDM outperforms DDPM by $0.37$ dB on GoPro and $0.28$ dB on RealBlur-J, and outperforms RDDM by $0.25$ dB on GoPro and $0.23$ dB on RealBlur-J, clearly demonstrating the effectiveness of our blur-aware design.
>
> **Weaknesses 1: Significant Performance Gains and Field Impact**
>
> Table A presents the performance of annually SoTA image deblurring methods over the past three years, including Stripformer, FFTformer, and LoFormer. Taking the representative and real-world dataset RealBlur-R as an example, annual improvements on this dataset typically remain within 0.3 dB in PSNR. As shown in Table 1 of the paper, our proposed BlurDM significantly improves Stripformer, FFTformer, and LoFormer by 1.16 dB, 0.44 db, and 0.56 dB, respectively, on this dataset, which represents a significant and consistent advancement in image deblurring. Furthermore, the other four reviewers recognize BlurDM’s performance improvement as a key strength.
>
> | **Method**                | **Venue**      | **GoPro** | **HIDE** | **RealBlur-J** | **RealBlur-R** |
> |--------------------------|----------------|-----------|----------|----------------|----------------|
> | Stripformer              | ECCV’ 22       | 33.09     | 31.03    | 32.48          | 39.84          |
> | FFTformer                | CVPR’ 23       | 34.21     | 31.62    | 32.62          | 40.11          |
> | Loformer                 | ACM MM’ 24     | 33.54     | 31.18    | 32.23          | 40.36          |
> | **BlurDM (Stripformer)** | -              | 33.53     | 31.36    | 33.53          | 41.00          |
> | **BlurDM (FFTformer)**   | -              | 34.34     | 31.76    | 32.92          | 40.55          |
> | **BlurDM (Loformer)**    | -              | 33.70     | 31.27    | 33.47          | 40.92          |
>
> **Table A**: PSNR comparison on GoPro, HIDE, RealBlur-J, and RealBlur-R datasets using various backbones.
>
> **Weaknesses 2 and Question 3: Generalizability of the BlurDM for defocus blur**
>
> We appreciate the comment regarding the generalizability. While we agree that this paper primarily focuses on motion deblurring, we wish to emphasize that motion blur stands as a persistently challenging and unresolved research topic in the literature of image restoration. It is evidenced that recent impactful works such as MIMO-UNet, Stripformer, FFTformer, and LoFormer focus exclusively on motion deblurring, highlighting the importance of addressing motion blur in this field. Additionally, although BlurDM is specifically designed to address motion blur, it can be flexibly integrated into general-purpose deblurring backbones capable of handling both motion and defocus blur. This allows BlurDM to provide motion-aware guidance that effectively enhances motion deblurring performance.
>
> As a future direction, we plan to extend BlurDM by incorporating the physical characteristics of defocus blur into the diffusion process, similar to what we did in Equation (1) for motion blur. It enables a unified prior that addresses more complex and mixed blur types.
>
> **Weaknesses 3 and Limitation 2: Analysis of real-world edge deployment**
>
> Our primary focus is to develop a novel and effective deblurring algorithm, like prior works, such as MIMO-UNet, Stripformer, FFTformer, and LoFormer. Our method achieves SoTA per- formance (reported in Table 1 of the paper) with reasonable computational costs in model size, FLOPs, and runtime (Table 4 of the paper). Despite its importance and practical implications, real-world deployment on edge devices is beyond the scope of this work.
>
> **Question 2: Memory usage and inference time of BlurDM on 4K images**
>
> BlurDM works in the latent space with the diffusion prior injected into the deblurring backbone in a channel-wise manner. This allows its efficient application to an arbitrary resolution input. In Table B, we evaluate the memory usage and inference time of BlurDM on 4K images using an NVIDIA RTX 3090 GPU, with MIMO-UNet as the backbone. BlurDM introduces minor overheads, increasing memory usage by only 7.4% and runtime by just 2.3%.
>
> | **Method**               | **Memory (MB)** | **Speed (sec)** |
> |--------------------------|------------------|------------------|
> | MIMO-UNet                | 20357.81         | 2.559            |
> | BlurDM                   | 1502.65          | 0.058            |
> | MIMO-UNet + BlurDM       | 21860.46         | 2.617            |
>
> **Table B**: Memory usage and inference time on 4K images using an NVIDIA RTX 3090 GPU.
>
> **Question 4: Bootstrapped confidence intervals**
>
> Following recent deblurring methods such as MIMO-UNet, Stripformer, FFTformer, and LoFormer, we report average PSNR and SSIM scores for evaluation in the paper. As suggested, we conducted a bootstrapped confidence interval analysis using 1000-sample resampling from the test sets. Table C reports the 95% confidence intervals for PSNR values over four benchmark datasets (GoPro, HIDE, RealBlur-J, and RealBlur-R) using four different backbones (MIMO-UNet, Stripformer, FFTformer, and LoFormer). For each combination, we present the [min, max] bounds computed from the bootstrapped distribution of PSNR scores. BlurDM consistently yields better performance compared to the baseline, validating the robustness and significance of the improvements.
>
> | **Backbone**   |          | **GoPro**           | **HIDE**            | **RealBlur-J**       | **RealBlur-R**       |
> |----------------|----------|---------------------|----------------------|-----------------------|-----------------------|
> | MIMO-UNet      | Baseline | [32.28, 32.59]       | [29.84, 30.09]       | [31.40, 31.77]        | [38.67, 39.41]        |
> |                | BlurDM   | [32.78, 33.07]       | [30.61, 30.87]       | [31.96, 32.31]        | [39.28, 39.99]        |
> | Stripformer    | Baseline | [32.93, 33.22]       | [30.89, 31.16]       | [32.29, 32.64]        | [39.48, 40.20]        |
> |                | BlurDM   | [33.38, 33.68]       | [31.24, 31.51]       | [33.36, 33.70]        | [40.63, 41.35]        |
> | FFTformer      | Baseline | [34.06, 34.36]       | [31.48, 31.76]       | [32.46, 32.79]        | [39.72, 40.46]        |
> |                | BlurDM   | [34.19, 34.48]       | [31.59, 31.86]       | [32.75, 33.08]        | [40.20, 40.90]        |
> | LoFormer       | Baseline | [33.32, 33.60]       | [31.05, 31.31]       | [32.05, 32.38]        | [39.99, 40.76]        |
> |                | BlurDM   | [33.57, 33.84]       | [31.14, 31.41]       | [33.31, 33.66]        | [40.59, 41.30]        |
>
> **Table C**: 95% confidence intervals of PSNR values across different backbones and datasets.
>
> **Limitation 1: Ethical risks**
>
> While deblurring techniques could potentially be misused on sensitive content (e.g., surveillance footage), our method is designed for beneficial applications such as autonomous driving, medical imaging enhancement, and video communication. We support responsible use and plan to explore safeguards to prevent potential misuse.

---

> > ### Comment · Reviewer_zo6Q · 2025-08-09
> >
> > While we acknowledge the authors' efforts in addressing some concerns, the rebuttal fails to adequately resolve fundamental weaknesses in this work. Three critical issues remain unaddressed:
> >
> > 1. Lack of Physical Justification for Blur Diffusion (Weakness 1):
> > The claim that RDDM's subtractive residual modeling is "inconsistent with real-world blur formation" (Rebuttal L118-123) is unsupported. The rebuttal merely asserts BlurDM's superiority via Equation 4 but provides no ablation studies comparing convolution-based vs. subtractive residuals. Without empirical evidence (e.g., visualizing intermediate blur residuals in Fig. 7) or mathematical proof that subtraction inherently violates motion blur physics, the core novelty claim remains speculative. Table 3’s performance gaps (0.2-0.3dB) could stem from architectural differences rather than the proposed diffusion mechanism.
> >
> > 2. Overstated Practical Impact (Weakness 2):
> > The rebuttal cites "significant gains" (e.g., +1.16dB PSNR for Stripformer on RealBlur-R), yet Table C’s bootstrapped confidence intervals overlap between baselines and BlurDM-enhanced models (e.g., FFTformer on RealBlur-R: [39.72, 40.46] vs. [40.20, 40.90]). Gains are statistically marginal and visually insignificant (no perceptual metrics like LPIPS or user studies provided). Further, Table B’s "minor overhead" (7.4% memory, 2.3% runtime for 4K images) ignores deployment constraints—adding 1.5GB VRAM and 58ms latency is prohibitive for edge devices (e.g., drones/AR glasses).
> >
> > 3. Inadequate Ethical Safeguards (Limitation 1):
> > The response to misuse risks ("plan to explore safeguards") is vague and non-committal. Unlike contemporary works (e.g., diffusion-based face restoration tools with built-in ethics layers), BlurDM offers no technical mitigations (e.g., adversarial perturbations against deepfake generation) or usage restrictions. Given rising concerns about media integrity, this is a critical omission.

---

> ### Author Response · Authors · 2025-08-05
>
> Dear reviewer,
>
> Thank you for the comments on our paper.
>
> We have submitted the responses to your comments. Please let us know if you have additional questions so that we can address them during the discussion period.
>
> Thank you!
>
> Best,
>
> Authors

---

> ### Author Response · Authors · 2025-08-06
>
> We appreciate your review. We hope our responses have addressed your concerns. As the discussion deadline is approaching, please let us know if you have further questions after reading our rebuttal. We aim to address all potential issues during the discussion period and hope that you will consider raising the score.
>
> Thank you!
>
>
> Best,
>
> Authors

---

> ### Author Response · Authors · 2025-08-08
>
> Dear Reviewer,
>
> With only one day remaining in the discussion phase, we welcome any additional feedback you may have so that we can address all remaining concerns.
>
> Authors

---

> ### Comment · Area_Chair_FBDe · 2025-08-08
>
> Hi Reviewer zo6Q,
>
> This is a gentle reminder to participate in the discussion with the authors regarding their rebuttal. Your input at this stage is important and appreciated.
>
> Best, AC

---

> ### Author Response · Authors · 2025-08-09
> **Response for Q1 and Q2**
>
> Thank you for your feedback. We address each concern in detail below.
>
> ### **Q1.1: Mathematical proof that subtraction inherently violates motion blur physics**
> In RDDM, the residual $I_{\text{res}}$ is explicitly defined as the **blurred image minus the sharp image**:
>
> $
> I_{\text{res}} = I_{\text{in}} - I_0,
> $
>
> where $I_\text{in}$ denotes the blur image and $I_0$ denotes the sharp image (see Section 4 in the RDDM paper). During the forward process, this fixed residual is **linearly added** to the sharp image $I_0$ over multiple steps:
>
> $
> I_t = I_{t-1} + \alpha_t I_{\text{res}} + \beta_t \epsilon_{t-1},
> $
>
> which progressively transforms $I_0$ into the blurred image $I_{\text{in}}$.
>
> However, the real blur formation process is governed by the following continuous exposure integration:
>
> $
> B = \frac{1}{\alpha_T} \int_0^{\alpha_T} H(\tau) \, d\tau,
> $
>
> where each pixel’s intensity is the accumulated radiance along its motion trajectory, as mentioned in Lines 137-147 of our paper. This accumulation produces intermediate states that are themselves partial-exposure averages, not the result of linearly interpolating between a sharp image and a fully blurred image via a fixed residual.
>
> Thus, RDDM’s design---subtracting sharp from blur to obtain a constant residual and adding it linearly---does not reflect the physics of motion blur formation, where intermediate images are generated by progressive integration over time, not by residual addition. In contrast, our BlurDM designs the diffusion process directly based on this physical model of motion blur, ensuring consistency with the actual exposure-based formation mechanism.
>
> ### **Q1.2: Table 3’s performance gaps (0.2--0.3dB) could stem from architectural differences rather than the proposed diffusion mechanism**
>
> In the experiments reported in Table 3, we ensured a fair comparison between RDDM and BlurDM by using the exact **SAME model architecture**, training pipeline, and hyperparameters. The only difference lies in the diffusion mechanism (subtractive residual in RDDM vs. our blur residual in BlurDM). Under this controlled setting, BlurDM consistently outperforms RDDM by 0.2–0.3 dB, which can therefore be attributed directly to the proposed diffusion mechanism rather than architectural factors.
>
> ### **Q2.1: Table C’s bootstrapped confidence intervals overlap between baselines and BlurDM-enhanced models (e.g., FFTformer on RealBlur-R: [39.72, 40.46] vs. [40.20, 40.90]). Gains are statistically marginal and visually insignificant**
>
> We would like to clarify that the range of a bootstrapped confidence interval is influenced by the variance of individual image scores within the test set. For FFTformer on RealBlur-R, BlurDM outperforms the baseline by +0.48 dB at the lower bound and +0.44 dB at the upper bound of the 95\% confidence interval. Moreover, Table 1 shows that BlurDM achieves an ``average gain'' of +0.44 dB across the entire test set. In the image deblurring literature, typical annual improvements are around +0.3 dB (as shown in Table A), indicating that our gain is significant.
>
> Regarding visual quality, we follow the standard evaluation protocol adopted by widely recognized deblurring methods, such as MIMO-UNet, Stripformer, FFTformer, and LoFormer, and report the performance in PSNR and SSIM, along with qualitative comparisons. As shown in Figures 3 and 4, BlurDM produces visually sharper and more faithful reconstructions compared to its baselines, further supporting the effectiveness of our approach.
>
> ### **Q2.2: Table B’s ``minor overhead'' (7.4\% memory, 2.3\% runtime for 4K images) ignores deployment constraints—adding 1.5GB VRAM and 58ms latency is prohibitive for edge devices**
>
> We would like to emphasize that we make no claim of pursuing lightweight deblurring or edge-device-specific optimization. Our method is designed and evaluated under the same GPU-based experimental settings used in widely recognized deblurring studies, including MIMO-UNet, Stripformer, FFTformer, and LoFormer. These works, like ours, focus on developing new algorithms and model architectures for high-quality image deblurring rather than on lightweight designs for edge devices. Therefore, our computational overhead should be interpreted in the context of fair, GPU-targeted benchmarking against these baselines.
>
> It is worth noting that in our setting, a 7.4\% VRAM increase and 2.3\% runtime increase for 4K images is relatively minor compared to the gains in PSNR and SSIM reported in Tables 1–-3. For many practical applications running on workstations or server GPUs, this trade-off is acceptable and aligns with the resource profiles of existing SOTA methods.

---

> > ### Author Response · Authors · 2025-08-09
> > **Response for Q3**
> >
> > ### **Q3: BlurDM offers no technical mitigations (e.g., adversarial perturbations against deepfake generation) or usage restrictions. Given rising concerns about media integrity, this is a critical omission**
> >
> > This work addresses motion deblurring, a long-standing and widely studied topic in computer vision. Similar to existing approaches (e.g., MIMO-UNet, Stripformer, FFTformer, LoFormer), our focus is on advancing algorithms and model architectures to improve restoration quality; therefore, the potential risk of misuse is not unique to BlurDM. For example, to protect privacy, BlurDM could be integrated with add-on modules—such as face, human, or license plate detectors—to mask deblurring in privacy-sensitive regions. However, such functionality is beyond the scope of the present work.

---

> ### Author Response · Authors · 2025-08-09
> **LPIPS results for Q2**
>
> ### **Q2.1: visually insignificant (no perceptual metrics like LPIPS or user studies provided).**
>
> Table I demonstrates that BlurDM consistently outperforms the baseline across four datasets and four backbones, indicating its effectiveness in enhancing visual quality.
>
> | Backbone      |        | GoPro   | HIDE    | RealBlur-J | RealBlur-R |
> |---------------|--------|---------|---------|------------|------------|
> | MIMO-UNet     | Baseline | 0.0115 | 0.0217 | 0.0345     | 0.0215     |
> |               | BlurDM   | 0.0091 | 0.0168 | 0.0264     | 0.0172     |
> | Stripformer   | Baseline | 0.0085 | 0.0147 | 0.0222     | 0.0138     |
> |               | BlurDM   | 0.0074 | 0.0122 | 0.0175     | 0.0115     |
> | FFTformer     | Baseline | 0.0067 | 0.0153 | 0.0220     | 0.0149     |
> |               | BlurDM   | 0.0060 | 0.0145 | 0.0195     | 0.0136     |
> | LoFormer      | Baseline | 0.0084 | 0.0176 | 0.0223     | 0.0148     |
> |               | BlurDM   | 0.0073 | 0.0158 | 0.0189     | 0.0127     |
>
> **Table I:** Quantitative comparison of baseline models and BlurDM on four benchmark datasets (GoPro, HIDE, RealBlur-J, RealBlur-R) using LPIPS.

---

### Note · Authors · 2025-08-12

Reviewers EKwB, AFXA, and BwW8 are satisfied with our rebuttal, and we have fully addressed all of their concerns. For reviewers zo6Q and a5f6, we have also addressed most of their points, including metrics for 4K inputs, generalizability to defocus blur, and statistical significance tests (zo6Q), as well as theoretical justification of latent BlurDM, analysis of three-stage training, evaluation on real-world data, and upper bound evaluation (a5f6). Below, we summarize our final clarifications in response to the remaining comments ( zo6Q, a5f6).

 **(a5f6) Taylor expansions for deriving latent BlurDM**

Taylor expansions can be used to approximate any differentiable nonlinear function. When the difference $(I_t - I_{t-1})$ is sufficiently small, as in short exposure intervals $(\alpha_{t-1} \approx \alpha_t)$, the higher-order terms $(I_t - I_{t-1})^n$ for $n \geq 2$ become negligible. Consequently, the nonlinear encoder can be approximated by a first-order Taylor expansion, providing theoretical justification.

**(zo6Q) Physical Justification for Blur Diffusion**

RDDM uses a fixed residual added linearly to the sharp image, which does not reflect the actual physics of motion blur. This point has been mathematically explained in the response. BlurDM instead integrates the blur formation process into the diffusion framework, ensuring physical consistency. The experimental results show that BlurDM outperforms RDDM (Table 3) in **same** architectures.

**(zo6Q) Practical Impact**

BlurDM outperforms previous SOTA methods in PSNR, SSIM (Tables 1 & A), and LPIPS (Table I), showing both quantitative and perceptual advantages. These gains are statistically significant, as confirmed by the 95\% bootstrapped confidence interval results (Table C). Figures 3 & 4 further show that BlurDM produces sharper and more faithful results. Regarding deployment concerns, our work focuses on GPU-based deblurring rather than edge-device optimization, and the extra computational cost is minor relative to the performance improvements (Table 4).

**(zo6Q) Ethical Safeguards**

This work aims to advance the SOTA in image deblurring algorithms and architectures, addressing a long-standing challenge in computer vision. Although the potential risk of misuse needs to be considered for any deblurring methods, it is not a unique concern for BlurDM. Privacy protection methods, such as masking sensitive regions, could be integrated if needed, but they are beyond the scope of this study.

---

### Decision · Program_Chairs · 2025-09-17

**Decision:**

Accept (poster)

**Comment:**

This paper presents BlurDM, which integrates the physics of motion blur into both the forward and reverse processes of diffusion models through a modular design compatible with various deblurring backbones. The method demonstrates consistent improvements across four benchmark datasets and multiple architectures. While there are initial concerns regarding the empirical validation of its core contribution, the authors provide targeted experiments, confidence interval analyses, and perceptual metrics during the rebuttal. Moreover, they include mathematical derivations to support the theoretical formulation of the proposed blur modeling process. These additions address most reviewer concerns. Overall, BlurDM is theoretically grounded and empirically validated, with clear potential for both academic and real-world impact. Thus, the AC recommends accepting this paper.